# Trust and vaccination intentions: Evidence from Lithuania during the COVID-19 pandemic

Laura Galdikiene[1], Jurate Jaraite[1], Agne Kajackaite[2,3]*

**1** Faculty of Economics and Business Administration, Vilnius University, Vilnius, Lithuania, **2** WZB Berlin Social Science Center, Berlin, Germany, **3** Department of Economics, Management and Quantitative Methods, University of Milan, Milan, Italy

\* agne.kajackaite@wzb.eu

## Abstract

In this paper, we study the relationship between trust and COVID-19 vaccination intentions. Vaccinating a large share of the population is essential for containing the COVID-19 pandemic. However, many individuals refuse to get vaccinated, which might be related to a lack of trust. Using unique survey data from Lithuania during the COVID-19 pandemic, we show that trust in government authorities, science, and pharmaceutical companies are important predictors of individual vaccination intentions. We do not find evidence that trust in strangers, the healthcare system, or the media predict intentions to get vaccinated against COVID-19.

**Data Availability Statement:** All relevant data are within the paper and its Supporting Information files.

## Introduction

### Motivation and research question

In 2020, the world was hit by the COVID-19 pandemic, which has become one of the worst health crises in human history [1]. Vaccines against COVID-19 were developed in record time, promising an effective solution to the pandemic [2]. Although the containment of the pandemic is in everyone's collective interest, some people have been reluctant to get vaccinated [3]. Hence, understanding the factors driving vaccination decisions is essential for designing effective policies and information campaigns to address current and future health crises. It could also help address other collective action problems (also referred to as social dilemmas), such as preventing the climate change disaster. Collective action problems are such situations, where individuals would be better off cooperating to achieve a common objective but fail to do so due to conflicting individual interests [4].

Previous findings in the literature have shown that trust may facilitate cooperation in the pursuit of socially valuable activities [5, 6]. Trust can also play an important role in situations with information asymmetries [7]. Since vaccination is related to a public good that requires cooperation and is characterized by information asymmetries between individuals and institutions involved in the vaccination process, in this paper, we aim to examine the association between different types of trust and willingness to get vaccinated. In particular, we study

**Funding:** The authors received no specific funding for this work.

**Competing interests:** The authors have declared that no competing interests exist.

whether trust in strangers (a concept that is closely related to generalized trust), government authorities, science, the healthcare system, pharmaceutical companies, and the media can predict individual vaccination intentions in the COVID-19 pandemic in Lithuania.

## Novelty of the study

Our study is novel in two aspects. First, we conduct a comprehensive analysis of the role of six different types of trust on vaccination intentions during the COVID-19 pandemic, while other studies similar to ours explore a single type of trust or at most a few selected types of trust [8–11]. To date, there is little evidence about the role of some specific types of trust, such as trust in strangers [7, 12] or trust in pharmaceutical companies, in explaining health behavior in a pandemic [13, 14].

The second novelty relates to our focus on Lithuania—an interesting and unexplored case in the pandemic literature. Lithuania has suffered greatly from the COVID-19 pandemic and it has faced sluggish vaccinations and high vaccine skepticism among the older population (see subsection "*COVID-19 and vaccinations in Lithuania*"). Lithuania is an interesting case also because of its post-Soviet legacy of relatively low levels of generalized trust, that is, general trust in other people, and trust in strangers [15], as well as trust in different institutions, including the Parliament, political parties, public administration, regional or local authorities, and health and medical staff [16].

## COVID-19 and vaccinations in Lithuania

In Lithuania, the first case of COVID-19 was confirmed on 28 February 2020. An immediate rapid rise in new COVID-19 cases was prevented, but in autumn of 2020 the pandemic situation took a turn for the worse and deteriorated further into the winter. Toward the end of 2020, Lithuania recorded more than 1,400 new daily COVID-19 cases per 1 million people—one of the worst results in the world at that time. In January 2021, when data for this study was collected, the curve of new infections was already going down, but the pandemic situation remained grim as the number of new cases, hospitalizations, and deaths from COVID-19 was still very high at that time [17].

Around the same time, the authorities started to administer vaccinations against COVID-19. The first doses of the approved COVID-19 vaccine (Comirnaty by Pfizer- BioNtech) were administered on 27 December 2020. At first, the limited vaccine resources were targeted toward key workers and clinically vulnerable groups. The vaccines were made available to the general public at the end of May 2021.

But despite the availability of vaccines, the pace of vaccinations remained sluggish throughout 2021. In the beginning of August 2021, before the restrictions on the unvaccinated were introduced, the vaccination rates in Lithuania stood below that of the European Union (EU) average [17]. Regarding age groups, Lithuania stood out for relatively low vaccination rates among its elderly population, which is not surprising as Lithuania is one of the EU countries with the highest COVID-19 vaccine skepticism in people over 50 [18].

Overall, the COVID-19 pandemic has cost many lives in Lithuania. The number of excess deaths, calculated as the deaths above the usual number of deaths that would have been observed under normal conditions, in Lithuania has been one of the largest globally. As of 12 June 2022, the estimated excess deaths per 100,000 persons stood at 745 in Lithuania, compared to 130 in Sweden [19]—one of the high-trust countries that has largely refrained from stringent pandemic containment measures and has vaccinated most of its elderly population [17, 20].

## Background and hypotheses

Many authors have examined individuals' voluntary vaccination decisions, which are often studied from the perspective of public goods theory [21, 22]. If vaccinations can stop the spread of infections, then such a containment of a virus is a public good, which requires people's cooperation in terms of them getting vaccinated. However, individuals personally have an incentive to free-ride and not incur the individual costs of vaccinations, such as safety concerns, potential side effects, costs of travel, and other monetary and non-monetary costs [23], while benefiting from the contained spread of the virus when a considerable number of people is vaccinated. In this way, free-riding can lead to a suboptimal collective outcome [24].

The social capital theory suggests that social capital, that is, certain shared values, norms, bonds, and trust among people, can help societies overcome the free-rider problem and facilitate cooperation in the pursuit of socially valuable activities [5, 6, 25–27]. When it comes to general trust in other people, that is, generalized trust, and, particularly, trust in strangers, more trusting individuals are more willing to cooperate and contribute to the public good, because they view other people as trustworthy and do not think they will be cheated [5]. In the context of vaccinations, this could mean that if people trust others, they are more willing to get vaccinated, because they do not think that others will free-ride and refuse vaccinations needed to stop the spread of the virus.

But empirical evidence on the role of social capital in shaping health behavior is somewhat conflicting. Some authors show that social capital (where trust is an important component) is associated with increased voluntary compliance to non-pharmaceutical interventions, including social distancing during the COVID-19 pandemic [28–31], as well as improved health outcomes, that is, fewer COVID-19 cases and fewer excess deaths per capita [32]. Some empirical studies also find a positive association between generalized trust and vaccination willingness [33–35]. However, some studies on health behavior find the opposite and do not confirm the theoretical considerations of the social capital theory. For example, Jennings et al. [36] show evidence of no significant relationship between generalized trust and vaccination intentions during the COVID-19 pandemic. Deopa and Fortunato [37] and Doganoglu and Ozdenoren [38] find a negative effect of generalized trust on social distancing behavior during the COVID-19 pandemic, arguing that when people trust others, they may believe that other people are sticking to restrictions and feel that it is safe to go out. Despite this conflicting pattern of results, we nevertheless expect that:

**Hypothesis 1:** *Trust in strangers predicts higher vaccination intentions (H1).*

Findings from the literature have shown that vaccinations can also depend on trust in various institutions and systems that produce and deliver vaccines as well as decide on their need [7]. In particular, this includes trust in pharmaceutical companies and science that develop the vaccines and ensure their safety and efficacy, trust in the healthcare system that administers vaccinations, and trust in policy-makers, mostly the government, that decide on the needed vaccine and establish the legal and regulatory framework for vaccinations [7, 10].

From the theoretical point of view, information asymmetry about the vaccine between individuals—less informed party—and institutions involved in the vaccination process—more informed party—makes trust in such institutions play an important role for the willingness to get vaccinated [7, 10]. Trust works as a heuristic shortcut to making a judgement by an individual with incomplete information about the risks and the benefits of vaccination, in particular, those related to a vaccine's safety and effectiveness, as well as its importance [10, 39, 40]. When individuals trust the institutions involved in the vaccination process, they believe that their representatives have the required competence and expertise, they have individuals' best interests at heart and adhere to the principles of integrity [7, 41].

Several empirical studies have confirmed the above theoretical consideration by providing evidence that there is a positive association between institutional trust and attitudes toward vaccination. For example, Jelnov and Jelnov [42] show that trust in the government leads to higher voluntary vaccination levels due to lower probability of a transparent and accountable government to promote an unsafe low-quality vaccine. Other authors have also found a positive association between trust in the government and willingness to get vaccinated [9, 33, 36, 41, 43, 44]. Similarly, trust in science or scientists [10, 33, 36, 41, 45], in the healthcare system or its workers [34, 45], and in pharmaceutical companies [13, 14] have been found to be positively related with vaccination intentions. Based on this literature, in our study, we raise the following hypotheses:

**Hypotheses 2–5:** *Trust in the government (H2)/ healthcare system (H3)/ science (H4)/ pharmaceutical companies (H5) predicts higher vaccination intentions.*

Meanwhile, the media is an important source of information about vaccination. The social learning theory suggests that trust in such information sources mediates the effect of exposure to information about a vaccine on attitudes toward vaccination [46]. That is, when the media provides people with information related to the vaccine, for example, outlines the benefits of vaccinations, and people believe the media to be a credible source of information, this information can have a positive effect on vaccine willingness. Different authors have also shown empirically that individuals' attitudes toward vaccinations are positively related to trust in the media [8, 46–48]. Based on this research, we will test whether:

**Hypothesis 6:** *Trust in the media predicts higher vaccination intentions (H6).*

## Present research

To study the relationship between trust and vaccination intentions in Lithuania, we conducted a representative online panel survey among Lithuanian adults in January 2021 (N = 1,000). In the survey, we collected information on individuals' intentions to get vaccinated against COVID-19 when such a free vaccine becomes available to them, six different types of trust, and many potential covariates.

We explore the following four types of trust in institutions that are involved in the vaccination process: trust in science; trust in pharmaceutical companies as these institutions together with scientific community develop the COVID-19 vaccines; trust in the healthcare system as it delivers the vaccines to individuals; and trust in government authorities, since they decide on the vaccine by approving it. We also study trust in the media, because it provides information on the vaccines and the vaccination process.

For interpersonal trust, we explore trust in strangers, because it should help societies overcome the free-riding problem and encourage cooperation between strangers [5, 6]. In this paper we focus on trust in strangers instead of the broader definition of generalized trust. The broader definition of generalized trust measures trust in other people in general, which is most often elicited by asking "Generally speaking would you say that most people can be trusted or that you can't be too careful in dealing with people? [49]". Generalized trust and trust in strangers are very closely related concepts and are often used as synonyms [5]. However, we think that trust in strangers eliminates the ambiguity inherent in the broader concept of generalized trust.

We find that higher trust in the government, science, and pharmaceutical companies is associated with a higher willingness to get vaccinated against COVID-19. We find no such evidence for trust in strangers, healthcare, or trust in the media. We also show that certain socio-demographic characteristics as well other factors, such as conspiracy beliefs, worries about COVID-19 and its effects, matter greatly for the intent to get vaccinated.

## Materials and methods

### Ethics statement

Ethical approval for data collection was issued by the WZB Research Ethics Committee. An informed written consent of participants was obtained.

### Survey

We employed a data set from a representative incentivized online panel survey conducted on 13–20 January 2021. We hired the company "Norstat" to implement the survey using its online access panel, that is, a group of registered internet users who have agreed to take part in various surveys. For participating in surveys, "Norstat" panel members are rewarded with virtual coins that could be exchanged into gift cards, coupons, or donated to a charity. The database of "Norstat" panel members was collected by the company by conducting member recruitment campaigns and representative surveys of the general population. The company sends individual invitations to potential panel members asking them to join the panel and individuals can then either accept or reject the invitations. The invitation-based system allows the company to ensure a diverse pool of individuals available for nationally representative surveys. Our survey participants were selected from the panel randomly according to the representativeness parameters, including age groups, gender, districts, and size of settlement (urban or rural). Invitations to participate in our survey were sent to potential participants by an automated system via email, which included a link to our questionnaire. The process of sending invitations continued until all sampling quotas for the target groups were fulfilled. The sampling quotas were set according to the population distribution data provided by Statistics Lithuania.

The informed written consent of participants was obtained by "Norstat" before individuals signed up to the panel. Individuals were informed that all their responses obtained in a survey will be anonymized. They were also told that their participation in a survey is entirely voluntary, and they are free to discontinue their participation at any time. To become members of the panel, individuals had to provide their consent by checking the "Yes, I agree to terms and conditions" box. On the first page of our questionnaire, we provided a description of our survey and its intent, the contact details of the research team, and asked the participants to contact us if they had any questions about the study. We also reminded the participants that their responses to the questionnaire will be anonymized. The survey would begin after the participant pressed "continue".

In the survey, we collected information on respondents' self-reported vaccination intentions during the COVID-19 pandemic, self-reported trust in different institutions, trust in the media, and trust in strangers. We also collected data on various beliefs and attitudes, as well as personal and demographic characteristics.

In total, 1,000 people aged 18 years and older answered the survey. It took an average of 11 minutes to answer the survey. Twenty-seven respondents who answered the survey in less than 4 minutes were dropped from the analysis. The full questionnaire can be found in S1 Text.

### Variables

The main outcome variable of this study is the intention to get vaccinated. In the survey, we asked the following question about the respondent's vaccination intent: "How much do you agree with this statement: I will get vaccinated as soon as a free COVID-19 vaccine becomes available to me." The respondents answered this question using a Likert scale ranging from 1 to 7, where 1 = "Strongly disagree" and 7 = "Strongly agree." Answers to this question were used to construct the dependent variable *vaccination*.

Our main explanatory variables are different types of trust, namely trust in strangers, government authorities, healthcare, science, the media, and pharmaceutical companies. To measure trust in strangers we ask respondents: "In general, how much do you trust people you do not know personally?" The question is answered on a 7-point scale ranging from 1 = "Do not trust at all" to 7 = "Trust completely." Our constructed question is similar to the trust in strangers question used in the World Values Survey (see [5]). Answers to this question measure variable *trust in strangers*.

To evaluate institutional trust and trust in the media, we ask respondents direct questions about their trust in specific institutions and the media: "In general, how much do you trust the country's government authorities/ the healthcare system/ science/ pharmaceutical companies/ the media?" The questions are again answered on a 7-point Likert scale. Answers to these questions measure the explanatory variables *trust in government*, *trust in healthcare*, *trust in science*, *trust in pharma*, and *trust in media*, respectively.

We also ask questions that could help control for other factors related to vaccination intentions. We want to control for individuals' and their relatives' health status, as poor health may increase COVID-19 risks and encourage vaccination. On the other hand, poor health status may be related to higher perceived risk of getting major side effects from vaccinations and may reduce willingness to get vaccinated. Thus, we ask respondents to answer questions about their personal (*personal health*) as well as the physical health of their close family members (*family health*). Regarding health, we also control for personal experience with COVID-19, that is, we ask whether individuals have been diagnosed with COVID-19 (*diagnosed with covid*) or they think they have been sick with COVID-19 (*think sick with covid*). People with such a disease history might have immunity against COVID-19, which may reduce the need for immediate vaccination against the disease.

Certain beliefs may also affect vaccination intentions. One of such factors is a belief in conspiracies, as previous studies have found that it is associated with reduced willingness to be vaccinated [45, 50–52]. We ask individuals, how much they agree with the statement, saying that the 5G mobile technology is directly related to the COVID-19 pandemic. Answers to this question are used to construct the variable *conspiracy beliefs*.

We also collect information on respondents' risk preferences. Findings in the literature have shown that risk aversion is mostly negatively associated with risky behavior in the health domain [53–55]. The variable *risk preferences* is constructed using responses to a question: "In general, I am willing to take risks [54]."

Data on COVID-19 related worries were also collected, such as self-reported worries of getting sick with COVID-19 (*fear of covid*) and a self-assessment of how the respondent's financial situation would be affected if the family's provider got sick with COVID-19 (*finances if sick*). These constructed variables measure the perceived susceptibility and perceived severity of the illness, which are two of the key constructs of the Health Belief Model [56] often used to predict health behavior. According to the model, if people regard themselves as susceptible to a certain medical condition or if they believe the condition could have serious consequences for them, they will be more willing to take action to prevent that medical condition from appearing.

We also collect data on respondents' sociodemographic characteristics, including age, gender, income, remote work possibilities, employment status, place of residence, size of the settlement, marital status, household size, education, and nationality. We treat all independent variables, constructed from answers to Liker-type questions, as continuous indices. In S1 Table we define our dependent and independent variables, as well as the control variables.

### Empirical specification

Our empirical strategy, which aims to explain the role of trust in vaccination intentions, proceeds as follows. First, we conduct a short descriptive analysis of the data to have a better understanding of the sample, the outcome variable, the main independent variables, and the strength of the relationships among the variables. Second, given the ordered nature of the responses to the vaccination question, we estimate an ordered logistic regression model using *vaccination* as the outcome variable.

We use three different specifications of the ordered logistic regression model. First, we start with regressing *vaccination* on every trust variable individually. This allows us to better understand the relationships between each trust variable and vaccination intent. We then regress *vaccination* on every trust variable (entered in separate regressions) and all control variables to control for socio-demographic characteristics of respondents, their health status, conspiracy beliefs, fears of getting sick with COVID-19, impact on finances in the case of COVID-19, and risk preferences. This specification provides a further check on the relationship between trust and vaccination intent and facilitates the evaluation of potential bias stemming from omitting other significant trust variables from the model. Finally, to test our hypotheses, we estimate a combined ordered logistic regression model, which includes all trust variables, and all control variables in a single regression. This third specification is our baseline model.

## Results and discussion

### Descriptive analysis

Twenty-seven respondents (out of 1,000) who answered the survey in less than 4 minutes were dropped from our analysis. This left us with the sample size of 973 observations. Around a half of these survey participants were women (55.4%) and almost half (48.2%) were aged 18–49. Lithuanians represented more than nine in ten (92.8%) of respondents. Seven in ten respondents had higher education (70.3%), were married or lived with a partner (70.9%), had household income lower than 2,000 euros (67.8%), and lived in a city or a town (67%). Almost one in three respondents lived in one of the three largest Lithuanian cities, that is, Vilnius (18%), Kaunas (9.5%), and Klaipeda (3.4%). Around one-fifth of participants had some personal experience with COVID-19, that is, they were either diagnosed with COVID-19 (7.3%) or they thought they have had COVID-19, but it has not been diagnosed (14.5%). S2 Table provides the additional characteristics of our data sample.

Most of respondents expressed a willingness to get vaccinated against COVID-19 (see Fig 1). In total, around 69% of respondents said that they strongly agree, agree, or agree somewhat to receive a vaccine as soon as it becomes available. Almost 19% of respondents expressed negative attitudes toward COVID-19 vaccines, that is, they answered that they strongly disagree, disagree, or disagree somewhat to get vaccinated. Out of these with negative attitudes, more than a half (10.6% of all respondents) disagreed strongly with getting vaccinated. Around 12% of respondents said they neither agree, nor disagree to get vaccinated and thus could be considered as undecided.

The intent to get vaccinated varies with the socio-demographic characteristics of respondents. One of such characteristics is age. Older participants of the survey were less skeptical about the vaccine and were more willing to get vaccinated than younger participants. Among those older than 60 years, 83% expressed an intent to get vaccinated, that is, they answered that they strongly agree, agree or agree somewhat to get vaccinated once the vaccine becomes available. Among those aged 40–59 years this number stood at 66% and at 58% among those younger than 39 years.

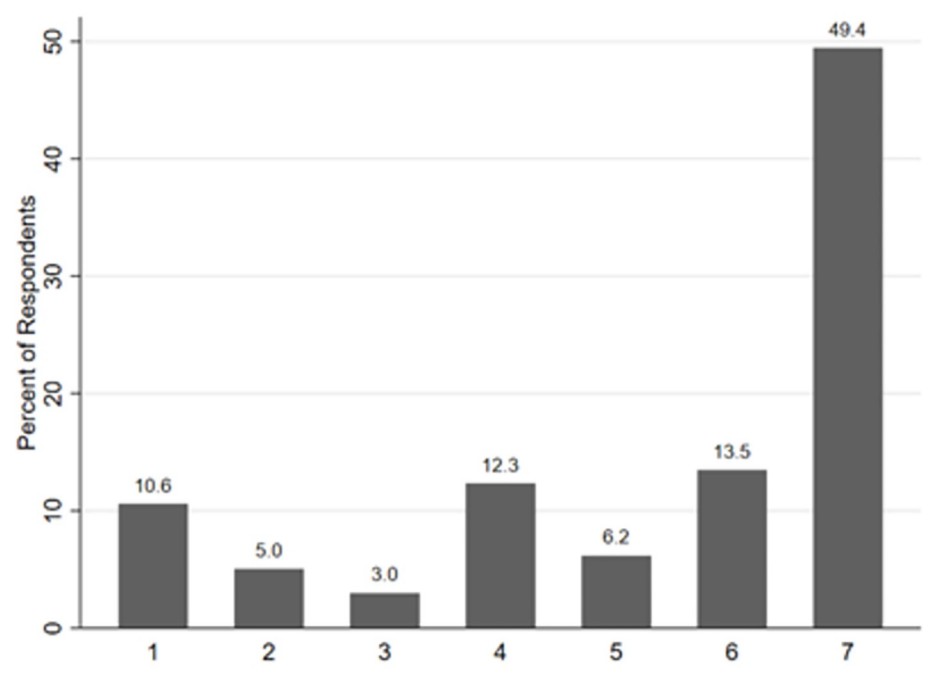

**Fig 1. Distribution of responses about vaccination intent.** *Note*: We asked respondents, how much they agree to the statement: "I will get vaccinated as soon as a free COVID-19 vaccine becomes available to me." Possible answers are 1 = "Strongly disagree," 2 = "Disagree," 3 = "Somewhat disagree," 4 = "Neither agree nor disagree," 5 = "Somewhat agree," 6 = "Agree," 7 = "Strongly agree." This figure shows the distribution of the responses to this statement.

In terms of gender, men had somewhat stronger positive views toward getting the vaccine than women—64% of men said that they agree or agree strongly to get vaccinated, while for women this number stood at 62%. Also, more women than men had strong negative attitudes toward the vaccine—16.5% of women disagreed strongly or disagreed with getting vaccinated, while among men 14.5% did. Among those respondents who were married or lived with a partner, somewhat more (70.3%) strongly agreed, agreed or agreed somewhat to get vaccinated as compared to those who were single or divorced (66%). Also, vaccination intent varied with household income. 79.4% of respondents with after-tax household income above 3,000 euros strongly agreed, agreed or agreed somewhat to get vaccinated, while among those with income lower than 500 euros only 60.3% did.

Regarding trust, respondents tended to trust institutions more than strangers. Around 23% expressed trust toward strangers, that is, answered that they trust completely, trust, or trust somewhat people they do not know personally (see Fig 2). For institutional trust, more than 47% of respondents answered that they trust government authorities, 48% trusted pharmaceutical companies, almost 55% trusted the healthcare system, and more than 84% of respondents trusted science. Only 38% of respondents expressed trust toward the media (see Fig 2). Here we consider that a person trusts an institution if he or she answered "trust completely," "trust," or "trust somewhat" to the trust questions in the survey. S3 Table provides the summary statistics of all variables used in the analysis.

S4 Table provides Spearman's correlation coefficients and their p-values for the outcome variable *vaccination* and all trust variables. It is evident that the relationship between vaccination intentions and the different trust variables is not equally strong. Correlations between vaccination intentions and different institutional trust variables, representing trust in the

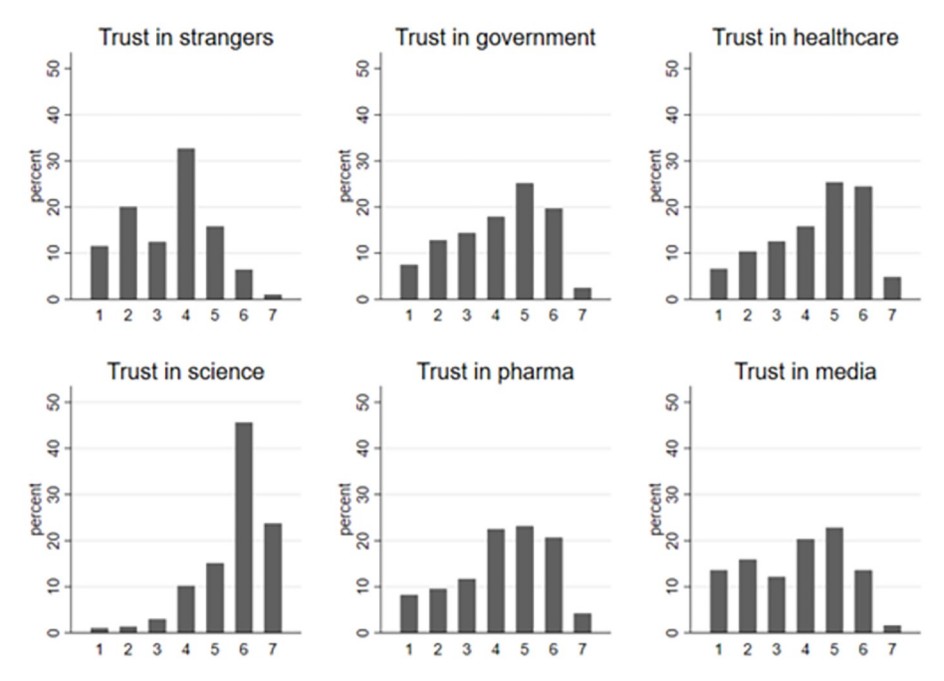

**Fig 2. Distribution of responses to trust questions.** *Note*: We ask respondents, how much they trust strangers, government authorities, the healthcare system, science, pharmaceutical companies, and the media. Possible answers are 1 = "Do not trust at all," 2 = "Do not trust," 3 = "Somewhat do not trust," 4 = "Neither trust nor distrust," 5 = "Somewhat trust," 6 = "Trust," 7 = "Trust completely." This figure shows the distributions of the responses to these questions.

government ($r_s = 0.426$, $p < 0.001$), trust in healthcare ($r_s = 0.387$, $p < 0.001$), trust in science ($r_s = 0.425$, $p < 0.001$), trust in pharmaceutical companies ($r_s = 0.409$, $p < 0.001$) and trust in the media ($r_s = 0.357$, $p < 0.001$), are statistically significant and positive. Correlation between *trust in strangers* and vaccination intentions ($r_s = 0.073$, $p = 0.023$) is rather weak, but still positive and statistically significant at a 5% significance level.

A Spearman's correlation was also run to assess the relationship among different trust variables. We find positive and statistically significant correlations among the trust variables, in particular, the institutional trust variables. For example, the correlation coefficient between trust in the government and trust in healthcare is 0.752 ($p < 0.001$) and between trust in the government and trust in the media is 0.558 ($p < 0.001$).

### Ordered logistic regression

**Main results.** In this subsection, we present and discuss the main results from estimating the ordered logistic regression models as described in subsection "*Empirical specification*." From each model we report the average marginal effects of trust variables.

First, we present the results from the simplest regression specification, which regresses *vaccination* on every trust variable in separate regressions without the control variables. As expected, we find the positive association between all trust variables and vaccination intentions. The logit coefficients are statistically significant at least at a 5% significance level (see S5 Table). The average marginal effects of all trust variables are also statistically significant at least at a 5% significance level for all categories of responses to the vaccination question (see columns 1.1–1.6 in Table 1). The average marginal effects of institutional trust variables are larger

**Table 1. Ordered logistic regression model with single trust variable and no controls.**

| Model | 1.1 | 1.2 | 1.3 | 1.4 | 1.5 | 1.6 |
|---|---|---|---|---|---|---|
| *Vaccination* | *Trust in strangers* | *Trust in government* | *Trust in healthcare* | *Trust in science* | *Trust in pharma* | *Trust in media* |
| Disagree strongly | -0.009** (0.004) | -0.051*** (0.005) | -0.044*** (0.005) | -0.058*** (0.006) | -0.048*** (0.005) | -0.040*** (0.005) |
| Disagree | -0.003** (0.002) | -0.018*** (0.003) | -0.016*** (0.002) | -0.021*** (0.003) | -0.018*** (0.002) | -0.015*** (0.002) |
| Disagree somewhat | -0.002** (0.001) | -0.009*** (0.002) | -0.008*** (0.002) | -0.011*** (0.002) | -0.009*** (0.002) | -0.007*** (0.001) |
| Neither agree, nor disagree | -0.006** (0.003) | -0.027*** (0.003) | -0.025*** (0.002) | -0.037*** (0.004) | -0.027*** (0.003) | -0.022*** (0.002) |
| Agree somewhat | -0.002** (0.001) | -0.008*** (0.001) | -0.008*** (0.001) | -0.013*** (0.002) | -0.008*** (0.001) | -0.007*** (0.001) |
| Agree | -0.001** (0.001) | -0.007*** (0.002) | -0.007*** (0.002) | -0.013*** (0.003) | -0.007*** (0.002) | -0.005*** (0.001) |
| Agree strongly | 0.022** (0.010) | 0.120*** (0.006) | 0.108*** (0.007) | 0.153*** (0.010) | 0.117*** (0.006) | 0.096*** (0.007) |
| Obs. | 973 | 973 | 973 | 973 | 973 | 973 |
| Pseudo R-sq. | 0.002 | 0.068 | 0.056 | 0.059 | 0.065 | 0.045 |
| LR chi-sq.(1) | 4.82** | 204.74*** | 167.12*** | 176.73*** | 194.44*** | 134.15*** |

*Note*: The table reports the average marginal effects obtained by estimating the ordered logit regression model with a single trust variable and no controls. The dependent variable is a 7-category variable *vaccination*. The first column of the table shows the responses to the vaccination question. Columns 1.1–1.6 show the results for estimating the different variations of the model where only the trust variable included in the model is changed. Standard errors are presented in parentheses below the average marginal effects. We also report the McFadden's pseudo R-squared (Pseudo R-sq.) and the Likelihood ratio chi-square statistic (LR chi-sq.). The Likelihood ratio test tests that at least one of the predictors' regression coefficients is not equal to zero in the model. The number in the parenthesis next to LR chi-sq. indicates the degrees of freedom of the chi-square distribution used to test the LR chi-square statistic and is defined by the number of predictors in the model.

*** $p < 0.01$

** $p < 0.05$

* $p < 0.1$.

in numerical terms than those of trust in strangers. Overall, the average marginal effects of trust in strangers are relatively small in numerical terms. On average, an increase in trust in strangers by 1 point is associated with a 2.2%-point greater probability of "agreeing strongly" to getting vaccinated. For trust in the government, the healthcare system, the pharmaceutical companies and the media, an increase in trust by 1 point is associated with around a 10–12%-point greater probability of reporting the highest vaccination intentions. Trust in science demonstrates the largest average marginal effects observed for the highest vaccination intentions (see column 1.4 in Table 1).

Next, we present the results from the regression models that regress vaccination intentions on every of the six trust variables separately and all control variables (sociodemographic characteristics of respondents, their health, conspiracy beliefs, fears of getting sick with COVID-19, impact on finances in the case of COVID-19, and risk preferences). We find that when we control for potential covariates, the logit coefficient of trust in strangers becomes statistically insignificant at any conventional significance level (see S6 Table). This is also reflected in the average marginal effects of trust in strangers, which are also statistically insignificant for all categories of responses to the vaccination question (see column 2.1 in Table 2). But the logit coefficients of the remaining trust variables, that is, trust in the government, trust in healthcare, trust in science, trust in pharma, and trust in the media, remain statistically significant at least at the 1% significance level (see S6 Table). The average marginal effects of these trust variables also keep their signs and remain statistically significant at the 1% level for all categories of vaccination variables (see columns 2.2–2.6 in Table 2). The association between vaccination intentions and the institutional trust variables remains positive. On average, an increase in trust in different institutions involved in the vaccination process by 1 point is associated with a 7.4–9.4%-point greater probability of agreeing strongly to getting vaccinated. Regarding the media, as trust in it goes up by 1 point, the probability of reporting highest vaccination intentions increases by 5.6%-points.

**Table 2. Ordered logistic regression model with single trust variable and all controls.**

| Model | 2.1 | 2.2 | 2.3 | 2.4 | 2.5 | 2.6 |
|---|---|---|---|---|---|---|
| *Vaccination* | *Trust in strangers* | *Trust in government* | *Trust in healthcare* | *Trust in science* | *Trust in pharma* | *Trust in media* |
| Disagree strongly | -0.005 (0.004) | -0.034*** (0.004) | -0.029*** (0.003) | -0.038*** (0.005) | -0.034*** (0.004) | -0.023*** (0.003) |
| Disagree | -0.002 (0.001) | -0.012*** (0.002) | -0.011*** (0.002) | -0.013*** (0.002) | -0.012*** (0.002) | -0.008*** (0.002) |
| Disagree somewhat | -0.001 (0.001) | -0.006*** (0.001) | -0.006*** (0.001) | -0.007*** (0.001) | -0.006*** (0.001) | -0.004*** (0.001) |
| Neither agree, nor disagree | -0.003 (0.002) | -0.019*** (0.002) | -0.017*** (0.002) | -0.022*** (0.003) | -0.020*** (0.002) | -0.013*** (0.002) |
| Agree somewhat | -0.001 (0.001) | -0.006*** (0.001) | -0.006*** (0.001) | -0.007*** (0.001) | -0.006*** (0.001) | -0.004*** (0.001) |
| Agree | -0.001 (0.001) | -0.006*** (0.001) | -0.006*** (0.001) | -0.008*** (0.002) | -0.006*** (0.001) | -0.004*** (0.001) |
| Agree strongly | 0.012 (0.009) | 0.083*** (0.007) | 0.074*** (0.007) | 0.094*** (0.010) | 0.085*** (0.007) | 0.056*** (0.007) |
| Obs. | 973 | 973 | 973 | 973 | 973 | 973 |
| Pseudo R-sq. | 0.124 | 0.160 | 0.155 | 0.148 | 0.164 | 0.141 |
| LR chi-sq.(31) | 372.01*** | 480.18*** | 464.23*** | 444.12*** | 493.01*** | 423.86*** |

*Note*: The table reports the average marginal effects obtained by estimating the ordered logit regression model with a single trust variable and all controls. We control for respondents' sociodemographic characteristics, health, conspiracy beliefs, fears of getting sick with COVID-19, impact on finances in the case of COVID-19, and risk preferences. The dependent variable is a 7-category variable *vaccination*. The first column of the table shows the responses to the vaccination question. Columns 2.1–2.6 show the results for estimating the different variations of the model where only the trust variable included in the model is changed. Standard errors are presented in parentheses below the average marginal effects. We also report the McFadden's pseudo R-squared (Pseudo R-sq.) and the Likelihood ratio chi-square statistic (LR chi-sq.). The Likelihood ratio test tests that at least one of the predictors' regression coefficients is not equal to zero in the model. The number in the parenthesis next to LR chi-sq. indicates the degrees of freedom of the chi-square distribution used to test the LR chi-square statistic and is defined by the number of predictors in the model.

*** $p < 0.01$

** $p < 0.05$

* $p < 0.1$.

However, the above-reported results from the first two regression models that include each trust variable separately could be misleading. That is, the estimated marginal effects of the trust variables may be biased upward, as the individually included trust variables might be capturing the effects of other trust variables that are omitted from the model. For this reason, the third regression specification, which includes all trust variables and controls, is estimated. This is our baseline model.

Estimating the baseline model yields logit coefficients that are statistically significant at the 1% significance level for trust in the government, trust in science, and trust in pharmaceutical companies (see S7 Table). The average marginal effects of these trust variables also remain statistically significant at the 1% level for all categories of vaccination variables. As trust in these institutions rises, the probability that the individuals report higher vaccination intentions increases (see Table 3). On average, an increase in trust in the government, science, and pharmaceutical companies by 1 point is associated with a 3.7, 3.9, and 4.7%-point greater probability of reporting the highest vaccination intentions, that is, "agreeing strongly" to getting vaccinated, respectively. In other words, higher trust in these institutions reduces the probability of disagreeing, being undecided, and agreeing less than strongly to getting vaccinated. These findings are in line with previous results in the literature [9, 10, 33, 43] and provide evidence in favor of hypotheses H2, H4, and H5.

Although some authors found that trust in healthcare [34, 45] and trust in the media [8, 46–48] were associated with higher vaccination intentions, in our case, these effects are potentially reduced by the inclusion of other trust variables, in particular, trust in the government and trust in science. The estimates of our baseline model show that the logit coefficients of trust in healthcare (H3) and trust in the media (H6) are statistically insignificant at any conventional level (see S7 Table). The average marginal effects of these trust variables are also statistically insignificant for all categories of responses to the vaccination question (see Table 3).

**Table 3. Ordered logistic regression model with all trust variables and controls.**

| Vaccination | Model 3 (baseline) | | | | | |
|---|---|---|---|---|---|---|
| | Trust in strangers | Trust in government | Trust in healthcare | Trust in science | Trust in pharma | Trust in media |
| Disagree strongly | 0.007** (0.003) | -0.015*** (0.005) | -0.004 (0.004) | -0.016*** (0.005) | -0.019*** (0.004) | -0.002 (0.003) |
| Disagree | 0.002* (0.001) | -0.005*** (0.002) | -0.002 (0.002) | -0.006*** (0.002) | -0.007*** (0.002) | -0.001 (0.001) |
| Disagree somewhat | 0.001* (0.001) | -0.003*** (0.001) | -0.001 (0.001) | -0.003*** (0.001) | -0.003*** (0.001) | 0.000 (0.001) |
| Neither agree, nor disagree | 0.004* (0.002) | -0.008*** (0.003) | -0.002 (0.003) | -0.009*** (0.003) | -0.011*** (0.002) | -0.001 (0.002) |
| Agree somewhat | 0.001* (0.001) | -0.003*** (0.001) | -0.001 (0.001) | -0.003*** (0.001) | -0.004*** (0.001) | 0.000 (0.001) |
| Agree | 0.001* (0.001) | -0.003*** (0.001) | -0.001 (0.001) | -0.003*** (0.001) | -0.004*** (0.001) | 0.000 (0.001) |
| Agree strongly | -0.017** (0.008) | 0.037*** (0.012) | 0.011 (0.011) | 0.039*** (0.011) | 0.047*** (0.009) | 0.005 (0.009) |
| Obs. | | | | | | 973 |
| Pseudo R-sq. | | | | | | 0.180 |
| LR chi-sq.(36) | | | | | | 540.25*** |

*Note*: The table reports the average marginal effects obtained by estimating the ordered logit regression model with all trust variables and all controls. We control for respondents' sociodemographic characteristics, health, conspiracy beliefs, fears of getting sick with COVID-19, impact on finances in the case of COVID-19, and risk preferences. The dependent variable is a 7-category variable *vaccination*. The first column of the table shows the responses to the vaccination question. Standard errors are presented in parentheses below the average marginal effects. We also report the McFadden's pseudo R-squared (Pseudo R-sq.) and the Likelihood ratio chi-square statistic (LR chi-sq.). The Likelihood ratio test tests that at least one of the predictors' regression coefficients is not equal to zero in the model. The number in the parenthesis next to LR chi-sq. indicates the degrees of freedom of the chi-square distribution used to test the LR chi-square statistic and is defined by the number of predictors in the model.

*** $p < 0.01$

** $p < 0.05$

* $p < 0.1$.

Furthermore, we find that when we estimate the baseline model, the logit coefficient of trust in strangers becomes statistically significant at the 5% significance level (see S7 Table). We also find that higher trust in strangers is associated with a lower probability of having high vaccination intentions (see the second column in Table 3). Thus, we do not find evidence in favor of H1. This result contrasts with the findings of some authors [33–35], who show that generalized trust is positively associated with the willingness to get vaccinated. One of the potential explanations of our result could be related to the fact that in this paper we focus on trust in strangers, which is a somewhat different concept than the broader concept of generalized trust analyzed in most other similar studies. Differences in the timing of surveys could also play a role—our survey was conducted quite early in the vaccination process, possibly before most people had internalized the social benefits of vaccinations. Another potential explanation for the negative association between trust in strangers and vaccinations is that when people trust others, they may believe that other people will protect against the disease, for example, by adhering to specialists' recommendations about safe health behavior during the pandemic and/ or by getting vaccinated, thus, they may feel safer about not rushing to get their vaccine. A few previous studies analyzing the role of generalized trust in explaining social distancing behavior during the COVID-19 pandemic have also found similar results [37, 38]. However, our estimated average marginal effects of trust in strangers are relatively small in numerical terms and are statistically significant at the 5% significance level only for the highest and lowest vaccination intentions (see the second column in Table 3), meaning that one should be cautious in drawing strong conclusions from this finding.

To evaluate if the baseline model (see Table 3) fits the data better than the reduced form models with single trust variables and all controls (see Table 2), we conducted likelihood ratio tests. The results show that adding all trust variables as predictor variables to a model, results

in a statistically significant improvement in the fit of the model (see S8 Table). Different diagnostic tests were also conducted for the baseline model. The tests did not detect model misspecification errors or problems of severe multicollinearity. The results of the tests can be obtained from the authors upon request.

In response to the request of one of the referees, as a robustness check, we estimate a multiple linear regression model with the same dependent and explanatory variables as in our baseline specification. The results from this model are largely in line with those obtained by estimating the baseline ordered logistic regression model as we find that trust in the government, science, and pharmaceutical companies is positively associated with vaccination intentions, while the coefficients on trust in healthcare and the media are statistically insignificant (see S9 Table). The coefficient on trust in strangers now is found to be statistically insignificant at any conventional significance level.

## Additional results

In this subsection, we report and discuss the additional findings obtained by estimating the baseline ordered logistic regression model. The control variables that we include in this model provide interesting insights about how individual characteristics, beliefs, and attitudes are associated with vaccination intentions. In S7 Table we report the logit coefficients of all control variables that are included in our baseline model. In S1 Fig we plot the average marginal effects of those control variables that have statistically significant (at least at a 5% significance level) logit coefficients.

We find that certain sociodemographic characteristics are significant predictors of vaccination intentions. The results demonstrate that, compared to men, women are less likely to have high intentions to get vaccinated. This finding is consistent with those by other authors [43, 57, 58]. Women could be more worried about the side effects of vaccines than men [58] or there could be differences in access to information about the vaccines between the two genders [43], which could lead to differences in their vaccination intentions.

Furthermore, we show that people from larger households are less likely to be in favor of vaccinations, which is somewhat related to the finding by Paul et al. [57], who showed that people living with children are less willing to get vaccinated. In addition, individuals who report having higher income as well as those who prefer not to answer the question about their income are more likely to have high vaccination intentions. Some authors have also found a positive association between income and vaccinations [9, 57]. We also find that individuals from Klaipeda—the third largest Lithuanian city and a major seaport with a relatively large Russian-speaking population—are less likely to be in favor of getting vaccinated.

When it comes to health, we find that people who think they were sick with COVID-19 but did not test for it are less likely to be willing to get vaccinated. Such people may have experienced mild COVID-19 symptoms, or they falsely believe that they have been sick with COVID-19, which makes them underestimate the threat of the disease or think that they are immune to contracting COVID-19. However, having been diagnosed with COVID-19 does not predict vaccination intentions.

We also find that beliefs in false information play a significant role in predicting vaccination intentions, which is in line with the findings of other studies [59–61]. Individuals who tend to believe in conspiracies, such as that 5G mobile technology is linked to the COVID-19 pandemic, are less likely to be in favor of vaccinating against COVID-19. Conspiracy beliefs can affect vaccination intentions negatively by reducing the perception of the threat of the virus and/ or by increasing the worries about the safety and the efficacy of vaccines [62].

Finally, as expected, the fear of getting sick with COVID-19 is associated with a higher probability of being in favor of vaccines. Similarly, if people think their finances would be

affected badly in the case their main family provider got sick with COVID-19 and could not work for some time, they are more likely to agree to take the vaccine.

## Conclusions

Our survey data show that the intent to get vaccinated is positively associated with trust in the government, science, and pharmaceutical companies. If such institutions are thought of as not being trustworthy, for example, because they are considered incompetent or corrupt, individuals are less likely to be in favor of getting vaccinated. Inherent mistrust in some of these institutions in Lithuania and some other countries could be a crucial factor contributing to the relatively low vaccination rate observed in 2021.

Although trust in strangers is an essential element of social capital [5], which should help societies prevent free-riding behavior, we find that it does not play a crucial role in predicting vaccination intent. One possible explanation for this is that some individuals, especially at the start of the vaccination process, may not view vaccinations as a way of contributing to the public good of containing the pandemic, that is, they do not yet internalize the social benefits of vaccinations against COVID-19. This could change at a later stage of the vaccination process as vaccinating against the virus is portrayed as a civic behavior intended to protect not only the person that is receiving the vaccine but also others. Thus, trust in strangers could still affect actual vaccination behavior as the vaccination campaign progresses. This conjecture is something future research could explore.

Our findings imply that societies that have more trusting relationships between individuals and key institutions, such as the government, science, and business, may contain the spread of the virus more rapidly and at a lower cost than societies where people trust institutions less. These institutions should recognize the role that trusting them plays in containing the COVID-19 pandemic and should take steps to build trust. However, building trust is unlikely to be easy or fast, as the COVID-19 pandemic itself, particularly if it is thought of as being mishandled, could have damaged trust [33, 63–65]. If trust does not improve, this could raise challenges for the management of health emergencies in the future as well as other crises that require collective action, such as the climate crisis. However, these questions are outside the scope of this study and are left for future research.

Apart from trust, our study also examined additional predictors of vaccination intentions. We find that women, individuals from larger households, and those who think they have been sick with COVID-19 are less likely to express willingness to receive a vaccine. We also show that misinformation plays a significant role in predicting vaccination intentions: individuals who tend to believe in conspiracies are less likely to agree to get vaccinated. Furthermore, higher expected personal financial costs of getting sick with COVID-19, higher income, and fear of getting sick with COVID-19 are associated with higher willingness to get vaccinated. These findings shed more light on the factors that are significant in predicting vaccinations. Targeting the vaccination campaign toward specific demographic groups of people who are less willing to get vaccinated, providing them with accurate information, and addressing their concerns could help increase vaccinations and contain the pandemic.

This study has some limitations. First, in the regressions, we do not control for the individuals' concerns about the safety, side-effects, and effectiveness of the vaccine against COVID-19, which are factors that have a significant effect on vaccination decisions. Thus, the estimates of the trust variables could partly capture the effects of these concerns. At the time of our survey there were only few personal experiences from the use of vaccines, thus some people could have had significant concerns about the safety of vaccines and their potential side effects. Second, the study investigates self-reported trust and vaccination intentions, which may suffer

from the social desirability bias and be prone to other concerns. The social desirability bias implies that survey respondents may overreport socially desirable and/ or underreport socially undesirable behavior. In our case, to be viewed favorably by others, respondents may indicate that they are more trusting and/ or more willing to get vaccinated than they actually are. But it is likely that this bias is not a very big concern here, as several studies have shown that the social desirability bias for self-reported health behaviors during the COVID pandemic, such as social distancing, is either very small [66] or even non-existent [67]. In addition, some studies have found that self-reported interpersonal and institutional trust predicts experimental trust measures [68–70]. However, there could be other factors than the social desirability bias that could lead to differences between vaccination intentions and the actual vaccine uptake, such as the temporal dynamics. Although in our study, many respondents expressed willingness to get vaccinated, which, according to Jensen et al. [71], should predict a high uptake of COVID-19 vaccines, the observed vaccination process was relatively slow in Lithuania. It may be the case that some individuals had second thoughts about getting vaccinated when the time to receive their vaccine came. To address the social desirability bias as well as other concerns related to the potential gap between vaccine willingness and actual vaccine uptake, future studies could use experimental measures of trust [70, 72] and actual vaccination behavior. Third, the evidence provided in the study is suggestive and we cannot draw causal inferences from it. The analysis of the causal relationship between trust and vaccination behavior is left for future research.

## Supporting information

**S1 Fig. Average marginal effects of control variables from the baseline model.** *Note*: The figures report the estimated average marginal effects of the control variables with 95% confidence intervals from the baseline ordered logistic regression model. The horizontal axis represents answers to the vaccination question: "I will get vaccinated as soon as a free COVID-19 vaccine becomes available to me." Answers range from 1 = "Strongly disagree" to 7 = "Strongly agree." We report average marginal effects only of those control variables that have logit coefficients that are statistically significant at least at a 5% significance level.
(PDF)

**S1 Table. Definition of variables.**
(PDF)

**S2 Table. Sample characteristics.**
(PDF)

**S3 Table. Summary statistics.**
(PDF)

**S4 Table. Correlation matrix of main variables.** *Note*: The table shows Spearman's correlation coefficients. P-values are provided in parentheses.
(PDF)

**S5 Table. Results of ordered logistic regression analysis with single trust variable and no controls.** *Note*: The table reports the logit coefficients obtained by estimating the ordered logit regression model with a single trust variable and no controls. The dependent variable is a 7-category variable *vaccination*. Standard errors are presented in parentheses below the coefficients. *** p < 0.01, ** p < 0.05, * p < 0.1.
(PDF)

**S6 Table. Results of ordered logistic regression analysis with single trust variable and all controls.** *Note*: The table reports the logit coefficients of trust variables obtained by estimating the ordered logit regression model with a single trust variable and all controls. We control for respondents' sociodemographic characteristics, health, conspiracy beliefs, fears of getting sick with COVID-19, impact on finances in the case of COVID-19, and risk preferences. The dependent variable is a 7-category variable *vaccination*. Standard errors are presented in parentheses below the coefficients. $p < 0.01$, ** $p < 0.05$, * $p < 0.1$.
(PDF)

**S7 Table. Results of ordered logistic regression analysis with all trust variables and controls.** *Note*: The table reports the logit coefficients obtained by estimating the ordered logit regression model with all trust variables and all controls (baseline specification). The dependent variable is a 7-category variable *vaccination*. Standard errors are presented in parentheses below the coefficients. *** $p < 0.01$, ** $p < 0.05$, * $p < 0.1$.
(PDF)

**S8 Table. Likelihood ratio tests.** *Note*: The table reports results from Likelihood ratio tests. The tests were conducted to evaluate the difference between nested models, that is, a more restrictive and a less restrictive model. The first column of the table shows which models are compared. LR chi-sq. gives the chi-square statistic for the likelihood ratio test. DF gives the degrees of freedom equal to the difference in the number of degrees of freedom between the two models that are compared.
(PDF)

**S9 Table. Results of multiple linear regression analysis.** *Note*: The table reports results obtained by estimating the baseline specification of our model using the OLS estimator. Variable *vaccination* is the dependent variable. It is regressed on all trust variables and all controls in a single model. We control for sociodemographic characteristics, health status, experience with COVID-19, conspiracy beliefs, fears of getting sick with COVID-19, impact on finances in the case of getting sick with COVID-19, and risk preferences. Robust standard errors are provided in parentheses *** $p < 0.01$, ** $p < 0.05$, * $p < 0.1$.
(PDF)

**S1 Text. Survey questionnaire.**
(PDF)

**S1 Dataset. Survey data.**
(XLSX)

## Acknowledgments

We thank Professor Karsten Staehr at Tallinn University of Technology, and Professor Mirko Moro at the University of Stirling for their useful comments and suggestions provided for the early draft of the paper. We are also grateful to the participants at Vilnius University's research seminar and the Baltic Economic Conference for their comments.

## Author Contributions

**Conceptualization:** Laura Galdikiene, Jurate Jaraite, Agne Kajackaite.

**Formal analysis:** Laura Galdikiene.

**Investigation:** Laura Galdikiene, Jurate Jaraite, Agne Kajackaite.

**Methodology:** Laura Galdikiene, Jurate Jaraite, Agne Kajackaite.

**Project administration:** Laura Galdikiene.

**Supervision:** Laura Galdikiene, Jurate Jaraite, Agne Kajackaite.

**Visualization:** Laura Galdikiene.

**Writing – original draft:** Laura Galdikiene.

**Writing – review & editing:** Laura Galdikiene, Jurate Jaraite, Agne Kajackaite.

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
