## [Decision Letter · Decision Letter 0]

15 Jun 2022

PONE-D-22-13409Trust and vaccination intentions: Evidence from Lithuania during the COVID-19 pandemicPLOS ONE

Dear Dr. Kajackaite,

Thank you for submitting your manuscript to PLOS ONE. After careful consideration, we feel that it has merit but does not fully meet PLOS ONE’s publication criteria as it currently stands. Therefore, we invite you to submit a revised version of the manuscript that addresses the points raised during the review process.

The manuscript requires further revisions with reference to the introductory section, empirical setting, along with concluding remarks. Also, the English language should be thoroughly corrected.

We look forward to receiving your revised manuscript.

Kind regards,

Stefan Cristian Gherghina, PhD. Habil.

Academic Editor

PLOS ONE

Journal Requirements:

Reviewers' comments:

Reviewer's Responses to Questions

**Comments to the Author**

1. Is the manuscript technically sound, and do the data support the conclusions?

Reviewer #1: Partly

Reviewer #2: Yes

2. Has the statistical analysis been performed appropriately and rigorously? 

Reviewer #1: No

Reviewer #2: Yes

3. Have the authors made all data underlying the findings in their manuscript fully available?

Reviewer #1: Yes

Reviewer #2: Yes

4. Is the manuscript presented in an intelligible fashion and written in standard English?

Reviewer #1: Yes

Reviewer #2: Yes

5. Review Comments to the Author

Reviewer #1: The research raises an interesting question by measuring and competing different types of trust to predict vaccination intentions. The sample is drawn from Lithuania, and is representative. The authors show a sense of perspective in interpreting their results, and the conclusion brings interesting issues to the table. I enjoyed reading this manuscript. However, I believe that several important issues should be addressed, both in terms of theoretical background and data analysis. Here are some detailed remarks:

Introduction

- In terms of structure, the introduction lacks clarity. I would perhaps separate more clearly the different part in the introduction, with (1) the section explaining the novelty, (2) the background, COVID vaccination in Lithuania (3) the variables and their links, (4) the hypotheses and (5) the “present research” section.

- Theoretical background is missing in general: there is little information on the state of the literature on variables on interest, or it is scattered throughout the manuscript. What results have been found on the different types of trusts in relation to vaccination? I would have expected a more extended introduction.

- In the same vein, I think that having a few more pieces of information about the context of vaccination against COVID in Lithuania would help the reader grasps the issues at stake.

- Why did you choose these trust variables? Institutional trust sometimes also includes trust in the police or the judicial system. The authors here are essentially looking at the different institutions involved in the COVID vaccination campaign. It would be worthwhile to explain this.

- In the introduction, authors mention the free-rider issue et and explain that generalized trust and trust in institutions can help diminish the free rider problem and therefore can predict vaccination intentions. If the free-rider issue is the mechanism behind the link between trust and vaccination intention, it should be detailed to a greater extent. Why would the free rider explain vaccination intention, and why would generalized trust reflect this? Why would institutional trust reflect this ? Particularly for institutional trust, it is not obvious that the free rider would explain the link between institutional trust and vaccination intention. Indeed, distrust in authorities is in fact related to distrust in the vaccine (as shown for example in Van Oost et al., 2022), thus a complete rejection and lack of trust in vaccine efficacy, not a free-rider ‘oportunistic’ behavior in which the free-rider believes that vaccination is efficient but does not want to pay its cost. In my opinion, the link between types of trust and vaccination intention should be justified to a much greater extent.

- Related to this, in the result section, authors cite litterature that show no link or a positive link of generalized trust on vaccine willingness. This literature contradicts the rationale suggested by the authors in the result section, who argue that a generalized trust could prevent individuals from getting vaccinated as people with higher levels of trust would perceive that others are reliable in their covid-risky behaviors. I believe that if authors stick to the free-rider rationale, they should includ such references in the introduction section.

- The authors briefly refer to “collective-action problems”. However, I do not think that the situation to which they refer is best described using this term of collective action (as defined by Becker or Van Zomeren). Perhaps authors should clarify the part where they refer to “collective-action problems”. In the same vein, authors refer to social capital, what do they mean here?

Methods

- In the survey section, I would mention more precisely how participants were contacted.

- I wonder why the authors aggregate participants' responses for vaccination intentions. Keeping a 7-level variable allows to exploit the variability in the participants' answers without having results that are harder to understand. Moreover, I find it problematic to analyze the results of the same item twice (after aggregating and without aggregating) without clear justification.

- About the trust variable, why are the two trust items used independently? The two trust items should theoretically measure the same concept and therefore be closely related, at least in a moderate way, but here the correlation is weak. This means that they reflect a different concept. This should be checked in the method section and addressed (by either aggregating the items, or separating them and theoretically justifying this choice). I believe that the low correlation between the two items of generalized trust might explained by the fact that the so-called "indirect" trust includes people who "live close to you" specifically, while the other item refers to trust in the general population. I would be more inclined to keep only the direct trust item.

- For control variables, one could also argue that poor health status (personal health) is related to higher perceived risk of getting major side effects.

Data analysis

- Why perform a logistic analysis when IV and DV are 7-level variables? I would have imagined using multiple regression analysis, which is also fairly straightforward, and is more appropriate for this type of data.

- In the descriptive analysis, considering that the data collection is representative, I would elaborate a bit more on the distribution of vaccination willingness in Lithuania.

- In the descriptive analysis, authors mention that “Both variables, representing trust in strangers, that is, trust strangers_d (rs = 0.073, p = 0.023) and trust strangers_i (rs = 0.103, p = 0.001) are not significantly correlated with vaccination intentions” while presenting p value that are under .05.

- Several time in the manuscript, author mention p =0.000, which is statistically impossible, and indicate a very small p-value close but not equal to 0.

- Authors indicate that the model with all control variables predicts vaccination intentions (DV) better than other models. However, adding a predictor in a model can only result in an improve in R-squared (a model with all control variables always has a greater Rsquared than a model with less of these control variables). Therefore, when comparing two models explaining the same DV, the question is whether the model with more predicting variables predicts significantly better than the other one. Is it the case here?

Conclusion

- I liked the conclusion section very much. I think that the reflections regarding trust and the crises in general (eg. climate crisis) were very relevant.

- I would disagree that the perceived side effects would not bias the results, due to the fact that the data collection occurred early in the vaccination campaign. Individuals might precisely be worried about side effects as very few people had been through, and documented the vaccine process yet.

Reviewer #2: Overall very interesting and well done. Just a few comments:

1. In the introduction you assert that social pressures probably played a small role when the survey was conducted, please provide more information or supporting research for this statement. I think in general adding more details on what the literature states on the relationship between social capital, trust, and health behaviors could be helpful if you have room. Some of this is in the discussion/conclusion and could be mentioned in the intro.

2. It is easier for the reader is you use a phrase to describe the variables in the text and tables instead of the variable name (for example "trust government" reads better as "trust in the government" and "trust stranger_i" reads better as "indirect measure of trust in strangers")

3. Please provide a new Table 1 where you provide information on the population (age, gender, race/ethnicity or nationality, responses to trust questions and all confounding variables) in the descriptive analysis section of the results. This is really important for the reader to better understand whose views are captured and many may not be aware of what the general demographics of Lithuania look like.

4. Some of the tables are hard to read and could be reduced to improve reader understanding. For example in the current Table 1, all covariates are included in all models and are mentioned in the table note, so they don't need to be in the table. Also this table has a lot of blank space, so this could be restructured so that each trust variable result is a column in a single row instead. Similarly, current Table 2 seems to include the main model and other models to assess the robustness of your results. Maybe moving the tables for the additional analyses to the supplementary material could help readability, so that the reader can focus on your main model.

5. The manuscript should also be reviewed to address errors in grammar. There are only a few of these, including one in the first sentence. It also needs to be edited for clarity in the results, when you're talking about how the variables were classified in the analysis. Also please make sure you follow the journal's guidelines for titling your sections!

6. Figures 1 and 2 use the 7-point Likert scale and then use the dichotomous version in the text of the results. Please be consistent in the presentation or remind the reader how they were categorized.

6. PLOS authors have the option to publish the peer review history of their article (what does this mean?). If published, this will include your full peer review and any attached files.

Reviewer #1: No

Reviewer #2: No

---

## [Author Response · Author response to Decision Letter 0]

25 Aug 2022

[OUR FULL RESPONSES DO NOT FIT INTO THIS SECTION. PLEASE SEE THE VERY END OF THE PDF FILE FOR OUR COMPLETE RESPONSES TO THE REVIEWERS. THANK YOU.]

Response to the Academic Editor

Dear Professor Gherghina ,

Thank you for giving us the opportunity to revise our paper for a possible publication in PLOS ONE. 

In the following, we document how we revised the paper in reaction to each of your and reviewers’ comments. Original comments by the review team are written in italics. 

1. Please ensure that your manuscript meets PLOS ONE’s style requirements, including those for file naming. 

Thank you for this comment. The manuscript has been reviewed to ensure that the manuscript is in line with journal’s style requirements.

Thank you for this comment. We have now addressed this comment by providing details on the participant consent and including the ethics statement in the manuscript and in online submission information. 

 In particular, we provide the ethics statement in the “Materials and methods” section of the manuscript on page 9: “Ethical approval for data collection was issued by the WZB Research Ethics Committee. An informed written consent of participants was obtained.”

 In the manuscript we also provide additional details on how the consent was obtained. On pages 9-10 we write: “The informed written consent of participants was obtained by “Norstat” before individuals signed up to the panel. Individuals were informed that all their responses obtained in a survey will be anonymized. They were also told that their participation in a survey is entirely voluntary, and they are free to discontinue their participation at any time. To become members of the panel, individuals had to provide their consent by checking the “Yes, I agree to terms and conditions” box. On the first page of our questionnaire, we provided a description of our survey and its intent, the contact details of the research team, and asked the participants to contact us if they had any questions about the study. We also reminded the participants that their responses to the questionnaire will be anonymized. The survey would begin after the participant pressed “continue”.”

3. In your Data Availability statement, you have not specified where the minimal data set underlying the results described in your manuscript can be found. PLOS defines a study’s minimal data set as the underlying data used to reach the conclusions drawn in the manuscript and any additional data required to replicate the reported study findings in their entirety. All PLOS journals require that the minimal data set be made fully available.

Thank you for this comment. We have now addressed it by uploading the minimal data set underlying the results described in the manuscript as a Supporting Information file, named “S1_Dataset.”

4. Please include your full ethics statement in the Methods section of your manuscript file. In your statement, please include the full name of the IRB or ethics committee who approved or waived your study, as well as whether or not you obtained informed written or verbal consent. If consent was waived for your study, please include this information in your statement as well.

Thank you for this comment. We have addressed this comment by providing the ethics statement in the “Materials and methods” section of the manuscript on page 9: “Ethical approval for data collection was issued by the WZB Research Ethics Committee. An informed written consent of participants was obtained.”

Thank you for your very helpful comments. We hope we addressed them in a satisfactory way. 

Response to Reviewer 1

Dear Reviewer,

 Thank you so much for reviewing our paper and for your thoughtful and constructive comments. Revising the paper based on your comments (in italics below) improved the paper significantly.

1. In terms of structure, the introduction lacks clarity. I would perhaps separate more clearly the different part in the introduction, with (1) the section explaining the novelty, (2) the background, COVID vaccination in Lithuania (3) the variables and their links, (4) the hypotheses and (5) the present research section.

Thank you so much for this useful comment. We agree that the introduction lacked clarity and structure. We have restructured the introduction following your recommendations. 

 Different parts in the introduction on pages 3-9 of the manuscript are now clearly separated using these headings: (1) “Motivation and research question”, (2) “Novelty of the study”, (3) “COVID-19 and vaccinations in Lithuania”, (4) “Background and hypotheses” (here we cover the theoretical background, literature on the variables of interest and their links to vaccinations as well as present the hypotheses. The hypotheses blended in naturally in this subsection, so this eliminated the need for a separate subsection), and (5) “Present research.”

2. Theoretical background is missing in general: there is little information on the state of the literature on variables on interest, or it is scattered throughout the manuscript. What results have been found on the different types of trusts in relation to vaccination? I would have expected a more extended introduction.

Thank you so much for this comment. We have now addressed your comment in a new subsection “Background and hypotheses” on pages 5-8. In this subsection we cover the theoretical background, literature on the variables of interest and their links to vaccinations and other health behaviors as well as outline the hypotheses. We write:

 “Many authors have examined individuals’ voluntary vaccination decisions, which are often studied from the perspective of public goods theory [21-22]. If vaccinations can stop the spread of infections, then such a containment of a virus is a public good, which requires people’s cooperation in terms of them getting vaccinated. However, individuals personally have an incentive to free-ride and not incur the individual costs of vaccinations, such as safety concerns, potential side effects, costs of travel, and other monetary and non-monetary costs [23], while benefiting from the contained spread of the virus when a considerable number of people is vaccinated. In this way, free-riding can lead to a suboptimal collective outcome [24]. 

 The social capital theory suggests that social capital, that is, certain shared values, norms, bonds, and trust among people, can help societies overcome the free-rider problem and facilitate cooperation in the pursuit of socially valuable activities [5-6, 25-27]. When it comes to general trust in other people, that is, generalized trust, and, particularly, trust in strangers, more trusting individuals are more willing to cooperate and contribute to the public good, because they view other people as trustworthy and do not think they will be cheated [5]. In the context of vaccinations this could mean that if people trust others, they are more willing to get vaccinated, because they do not think that others will free-ride and refuse vaccinations needed to stop the spread of the virus.

 Different authors have shown social capital to be positively associated with health behavior, such as voluntary compliance to non-pharmaceutical interventions, including social distancing, during the COVID-19 pandemic [28-32]. Some empirical studies have also found a positive association between generalized trust and vaccination willingness [33-35], but some find no significant relationship between this type of trust and vaccination intentions [36]. A few previous studies have found a negative effect of generalized trust on social distancing behavior during the COVID-19 pandemic, arguing that when people trust others, they may believe that other people are sticking to restrictions and feel that it is safe to go out during the pandemic [37-38]. Hence, based on the social capital theory and previous empirical literature on trust and vaccinations, we hypothesize that:

 Hypothesis 1: Trust in strangers predicts higher vaccination intentions (H1).

 Findings from the literature have shown that vaccinations can also depend on trust in various institutions and systems that produce and deliver vaccines as well as decide on their need [7]. In particular, this includes trust in pharmaceutical companies and science that develop the vaccines and ensure their safety and efficacy, trust in the healthcare system that administers vaccinations, and trust in policy-makers, mostly the government, that decide on the needed vaccine and establish the legal and regulatory framework for vaccinations [7, 10]. 

 From the theoretical point of view, information asymmetry about the vaccine between individuals—less informed party—and institutions involved in the vaccination process—more informed party—makes trust in such institutions play an important role for the willingness to get vaccinated [7, 10]. Trust in institutions helps individuals with incomplete information weight the risks and benefits of vaccinations [39]. In this way, trust works as a heuristic shortcut to making a judgement by an individual with incomplete information about the safety, effectiveness, and importance of a vaccine in question [10, 40]. When individuals trust the institutions involved in the vaccination process, they believe that their representatives have the required competence and expertise, they have individuals’ best interests at heart and adhere to the principles of integrity [7, 41]. 

 Several empirical studies have confirmed the above theoretical consideration by providing evidence that there is a positive association between institutional trust and attitudes toward vaccination. For example, Jelnov and Jelnov [42] show that trust in the government leads to higher voluntary vaccination levels due to lower probability of a transparent and accountable government to promote an unsafe low-quality vaccine. Other authors have also found a positive association between trust in the government and willingness to get vaccinated [9, 33, 36, 41, 43-44]. Similarly, trust in science or scientists [10, 33, 36, 41, 45], in the healthcare system or its workers [34, 45-46], and in pharmaceutical companies [13-14] have been found to be positively related with vaccination intentions. Based on this literature, in our study, we raise the following hypotheses:

 Hypotheses 2-5: Trust in the government (H2)/ healthcare system (H3)/ science (H4)/ pharmaceutical companies (H5) predicts higher vaccination intentions.

 Meanwhile, the media is an important source of information about vaccination. The social learning theory suggests that trust in such information sources mediates the effect of exposure to information about a vaccine on attitudes toward vaccination [47]. That is, when the media provides people with information related to the vaccine, for example, outlines the benefits of vaccinations, and people believe the media to be a credible source of information, this information can have a positive effect on vaccine willingness. Different authors have also shown empirically that individuals’ attitudes toward vaccinations are positively related to trust in the media [8, 47-48]. Based on this research, we will test whether:

 Hypothesis 6: Trust in the media predicts higher vaccination intentions (H6).”

3. In the same vein, I think that having a few more pieces of information about the context of vaccination against COVID in Lithuania would help the reader grasps the issues at stake.

Thank you for the comment. We now describe the Lithuanian context of the COVID-19 pandemic, including vaccinations, in our new “COVID-19 and vaccinations in Lithuania” subsection on pages 4-5:

 “In Lithuania, the first case of COVID-19 was confirmed on 28 February 2020. An immediate rapid rise in new COVID-19 cases was prevented, but in autumn of 2020 the pandemic situation took a turn for the worse and deteriorated further into the winter. Toward the end of 2020, Lithuania recorded more than 1,400 new daily COVID-19 cases per 1 million people—one of the worst results in the world at that time. In January 2021, when data for this study was collected, the curve of new infections was already going down, but the pandemic situation remained grim as the number of new cases, hospitalizations, and deaths from COVID-19 was still very high at that time [17].

 Around the same time, the authorities started to administer vaccinations against COVID-19. The first doses of the approved COVID-19 vaccine (Comirnaty by Pfizer- BioNtech) were administered on 27 December 2020. At first, the limited vaccine resources were targeted toward key workers and clinically vulnerable groups. The vaccines were made available to the general public at the end of May 2021.

 But despite the availability of vaccines, the pace of vaccinations remained sluggish throughout 2021. In the beginning of August 2021, before the restrictions on the unvaccinated were introduced, the vaccination rates in Lithuania stood below that of the European Union (EU) average [17]. Regarding age groups, Lithuania stood out for relatively low vaccination rates among its elderly population, which is not surprising as Lithuania is one of the EU countries with the highest COVID-19 vaccine skepticism in people over 50 [18].

 Overall, the COVID-19 pandemic has cost many lives in Lithuania. The number of excess deaths, calculated as the deaths above the usual number of deaths that would have been observed under normal conditions, in Lithuania has been one of the largest globally. As of 12 June 2022, the estimated excess deaths per 100,000 persons stood at 745 in Lithuania, compared to 130 in Sweden [19]—one of the high-trust countries that has largely refrained from stringent pandemic containment measures and has vaccinated most of its elderly population [17, 20].”

4. Why did you choose these trust variables? Institutional trust sometimes also includes trust in the police or the judicial system. The authors here are essentially looking at the different institutions involved in the COVID vaccination campaign. It would be worthwhile to explain this.

Thank you very much for the comment. We now explain the choice of trust variables in the subsection “Background and hypotheses” on pages 5-8 (please see the response to your comment No. 2). We also summarize the rationale for choosing the specific trust variables in subsection “Present research”. In particular, on pages 8-9 we write: 

 “We explore the following four types of trust in institutions that are involved in the vaccination process: trust in science; trust in pharmaceutical companies as these institutions together with scientific community develop the COVID-19 vaccines; trust in the healthcare system as it delivers the vaccines to individuals; and trust in government authorities, since they decide on the vaccine by approving it. We also study trust in the media, because it provides information on the vaccines and the vaccination process. 

 For interpersonal trust, we explore trust in strangers, because it should help societies overcome the free-riding problem and encourage cooperation between strangers [5-6]. In this paper we focus on trust in strangers instead of generalized trust, which is measured using the broader trust question that asks about trust in other people in general [49]. Although the two types of trust are very closely related concepts [5], we think that trust in strangers eliminates the ambiguity inherent in the broader concept of generalized trust.”

5. In the introduction, authors mention the free-rider issue et and explain that generalized trust and trust in institutions can help diminish the free rider problem and therefore can predict vaccination intentions. If the free-rider issue is the mechanism behind the link between trust and vaccination intention, it should be detailed to a greater extent. Why would the free rider explain vaccination intention, and why would generalized trust reflect this? Why would institutional trust reflect this? Particularly for institutional trust, it is not obvious that the free rider would explain the link between institutional trust and vaccination intention. Indeed, distrust in authorities is in fact related to distrust in the vaccine (as shown for example in Van Oost et al., 2022), thus a complete rejection and lack of trust in vaccine efficacy, not a free-rider “opportunistic” behavior in which the free-rider believes that vaccination is efficient but does not want to pay its cost. In my opinion, the link between types of trust and vaccination intention should be justified to a much greater extent.

Thank you for the comment. We have now added a discussion of the link between different types of trust and vaccination intentions in our “Background and hypotheses” subsection.

 In particular, regarding the link between generalized trust and vaccination intention, on pages 5-6, we write: 

 “Many authors have examined individuals’ voluntary vaccination decisions, which are often studied from the perspective of public goods theory [21-22]. If vaccinations can stop the spread of infections, then such a containment of a virus is a public good, which requires people’s cooperation in terms of them getting vaccinated. However, individuals personally have an incentive to free-ride and not incur the individual costs of vaccinations, such as safety concerns, potential side effects, costs of travel, and other monetary and non-monetary costs [23], while benefiting from the contained spread of the virus when a considerable number of people is vaccinated. In this way, free-riding can lead to a suboptimal collective outcome [24]. 

 The social capital theory suggests that social capital, that is, certain shared values, norms, bonds, and trust among people, can help societies overcome the free-rider problem and facilitate cooperation in the pursuit of socially valuable activities [5-6, 25-27]. When it comes to general trust in other people, that is, generalized trust, and, particularly, trust in strangers, more trusting individuals are more willing to cooperate and contribute to the public good, because they view other people as trustworthy and do not think they will be cheated [5]. In the context of vaccinations this could mean that if people trust others, they are more willing to get vaccinated, because they do not think that others will free-ride and refuse vaccinations needed to stop the spread of the virus.

 Different authors have shown social capital to be positively associated with health behavior, such as voluntary compliance to non-pharmaceutical interventions, including social distancing, during the COVID-19 pandemic [28-32]. Some empirical studies have also found a positive association between generalized trust and vaccination willingness [33-35], but some find no significant relationship between this type of trust and vaccination intentions [36]. A few previous studies have found a negative effect of generalized trust on social distancing behavior during the COVID-19 pandemic, arguing that when people trust others, they may believe that other people are sticking to restrictions and feel that it is safe to go out during the pandemic [37-38].”

 Regarding the link between institutional trust and vaccination intentions, on pages 6-8, we write: 

 “Findings from the literature have shown that vaccinations can also depend on trust in various institutions and systems that produce and deliver vaccines as well as decide on their need [7]. In particular, this includes trust in pharmaceutical companies and science that develop the vaccines and ensure their safety and efficacy, trust in the healthcare system that administers vaccinations, and trust in policy-makers, mostly the government, that decide on the needed vaccine and establish the legal and regulatory framework for vaccinations [7, 10]. 

 From the theoretical point of view, information asymmetry about the vaccine between individuals—less informed party—and institutions involved in the vaccination process—more informed party—makes trust in such institutions play an important role for the willingness to get vaccinated [7, 10]. Trust in institutions helps individuals with incomplete information weight the risks and benefits of vaccinations [39]. In this way, trust works as a heuristic shortcut to making a judgement by an individual with incomplete information about the safety, effectiveness, and importance of a vaccine in question [10, 40]. When individuals trust the institutions involved in the vaccination process, they believe that their representatives have the required competence and expertise, they have individuals’ best interests at heart and adhere to the principles of integrity [7, 41]. 

 Several empirical studies have confirmed the above theoretical consideration by providing evidence that there is a positive association between institutional trust and attitudes toward vaccination. For example, Jelnov and Jelnov [42] show that trust in the government leads to higher voluntary vaccination levels due to lower probability of a transparent and accountable government to promote an unsafe low-quality vaccine. Other authors have also found a positive association between trust in the government and willingness to get vaccinated [9, 33, 36, 41, 43-44]. Similarly, trust in science or scientists [10, 33, 36, 41, 45], in the healthcare system or its workers [34, 45-46], and in pharmaceutical companies [13-14] have been found to be positively related with vaccination intentions. (…)

 Meanwhile, the media is an important source of information about vaccination. The social learning theory suggests that trust in such information sources mediates the effect of exposure to information about a vaccine on attitudes toward vaccination [47]. That is, when the media provides people with information related to the vaccine, for example, outlines the benefits of vaccinations, and people believe the media to be a credible source of information, this information can have a positive effect on vaccine willingness. Different authors have also shown empirically that individuals’ attitudes toward vaccinations are positively related to trust in the media [8, 47-48].”

6. Related to this, in the result section, authors cite literature that show no link or a positive link of generalized trust on vaccine willingness. This literature contradicts the rationale suggested by the authors in the result section, who argue that a generalized trust could prevent individuals from getting vaccinated as people with higher levels of trust would perceive that others are reliable in their covid-risky behaviors. I believe that if authors stick to the free-rider rationale, they should include such references in the introduction section.

Thank you for the comment. We have now added references on the negative association between generalized trust and health behavior in the “Background and hypotheses” subsection. In particular, on page 6 we write: “A few previous studies have found a negative effect of generalized trust on social distancing behavior during the COVID-19 pandemic, arguing that when people trust others, they may believe that other people are sticking to restrictions and feel that it is safe to go out during the pandemic [37-38].”

7. The authors briefly refer to “collective-action problems”. However, I do not think that the situation to which they refer is best described using this term of collective action (as defined by Becker or Van Zomeren). Perhaps authors should clarify the part where they refer to “collective-action problems”. In the same vein, authors refer to social capital, what do they mean here?

Thank you for the comment. To clarify the term of “collective action problem” we include an explanation on page 3. In particular, we write: “Hence, understanding the factors driving vaccination decisions is essential for designing effective policies and information campaigns to address current and future health crises. It could also help address other collective action problems (also referred to as social dilemmas), such as preventing the climate change disaster. Collective action problems are such situations, where individuals would be better off cooperating to achieve a common objective but fail to do so due to conflicting individual interests [4].”

 To clarify the “social capital” term, on page 6, we write: “The social capital theory suggests that social capital, that is, certain shared values, norms, bonds, and trust among people, can help societies overcome the free-rider problem and facilitate cooperation in the pursuit of socially valuable activities [5-6, 25-27].”

8. In the survey section, I would mention more precisely how participants were contacted.

Thank you so much for this comment. We have now included more details on how the survey participants were contacted. In the “Survey” subsection, on page 9, we write: “We hired the company “Norstat” to implement the survey using its online access panel, that is, a group of registered internet users who have agreed to take part in various surveys. For participating in surveys, “Norstat” panel members are rewarded with virtual coins that could be exchanged into gift cards, coupons, or donated to a charity. Our survey participants were selected from the panel randomly, and invitations to participate in our survey were sent to them by an automated system via email, which included a link to our questionnaire.”

9. I wonder why the authors aggregate participants’ responses for vaccination intentions. Keeping a 7-level variable allows to exploit the variability in the participants’ answers without having results that are harder to understand. Moreover, I find it problematic to analyze the results of the same item twice (after aggregating and without aggregating) without clear justification.

Thank you so much for this useful comment. We agree that keeping the 7-category variable allows us to exploit the variability in the participants’ answers. Following your comment, we decided not to aggregate the responses for vaccination intentions. As a result, we removed the binary logistic regression model that uses the variable provaccine, that is, the dichotomized version of vaccination intentions, as the dependent variable. In the manuscript, we now focus on the results obtained from the ordered logistic regression analysis that uses the 7-level variable for vaccination intentions (vaccination) as the dependent variable. We explore three specifications of the ordered logistic regression model. We explain our empirical strategy in the “Empirical specification” subsection, on pages 12-13. In particular, we write:

 “Our empirical strategy, which aims to explain the role of trust in vaccination intentions, proceeds as follows. First, we conduct a short descriptive analysis of the data to have a better understanding of the sample, the outcome variable, the main independent variables, and the strength of the relationships among the variables. Second, given the ordered nature of the responses to the vaccination question, we estimate an ordered logistic regression model using vaccination as the outcome variable. 

 We use three different specifications of the ordered logistic regression model. First, we start with regressing vaccination on every trust variable individually. This allows us to better understand the relationships between each trust variable and vaccination intent. We then regress vaccination on every trust variable (entered in separate regressions) and all control variables to control for socio-demographic characteristics of respondents, their health status, conspiracy beliefs, fears of getting sick with COVID-19, impact on finances in the case of COVID-19, and risk preferences. This specification provides a further check on the relationship between trust and vaccination intent and facilitates the evaluation of potential bias stemming from omitting other significant trust variables from the model. Finally, to test our hypotheses, we estimate a combined ordered logistic regression model, which includes all trust variables, and all control variables in a single regression. This third specification is our baseline model.” 

 Since we remove the analysis of the variable with aggregated responses on vaccination intentions, we now describe our results only in subsection “Ordered logistic regression.”

10. About the trust variable, why are the two trust items used independently? The two trust items should theoretically measure the same concept and therefore be closely related, at least in a moderate way, but here the correlation is weak. This means that they reflect a different concept. This should be checked in the method section and addressed (by either aggregating the items, or separating them and theoretically justifying this choice). I believe that the low correlation between the two items of generalized trust might explained by the fact that the so-called "indirect" trust includes people who "live close to you" specifically, while the other item refers to trust in the general population. I would be more inclined to keep only the direct trust item.

Thank you so much for this comment. Given the weak correlation between the direct and indirect measures of trust, we decided to remove the indirect trust measure from the analysis and keep only the direct trust item.

11. For control variables, one could also argue that poor health status (personal health) is related to higher perceived risk of getting major side effects.

Thank you so much for this comment. We agree that poor health status could be related to higher perceived risk of getting major side effect from vaccination. On page 11 we write: “We want to control for individuals’ and their relatives’ health status, as poor health may increase COVID-19 risks and encourage vaccination. On the other hand, poor health status may be related to higher perceived risk of getting major side effects from vaccinations and may reduce willingness to get vaccinated.”

12. Why perform a logistic analysis when IV and DV are 7-level variables? I would have imagined using multiple regression analysis, which is also fairly straightforward, and is more appropriate for this type of data.

 Thank you so much for this comment. Our dependent variable (vaccination) is constructed from an individual Likert-type question and is thus generally considered an ordinal dependent variable. Although some authors argue that ordinary linear model techniques can be applied to ordinal variables (e.g., Norman, 2010), others suggest following a more cautious approach and abstain from linear regression analysis with ordinal dependent variables (e.g., Liddell and Kruschke, 2018; McKelvey and Zavoina, 1975; Winship and Mare, 1984). Although the categories for our ordinal dependent variable are coded as consecutive integers from 1 to 7, we cannot assume that the distances between categories are equal. That is, we cannot assume that the distance between strongly agreeing and agreeing is the same as the distance between agreeing somewhat and neither agreeing, nor disagreeing (Jamieson, 2005; Scott Long and Freese, 2014). Thus, an ordinal dependent variable violates the assumptions of the linear regression model and some authors show that the use of this model can lead to incorrect conclusions (Liddell and Kruschke, 2018; McKelvey and Zavoina, 1975; Winship and Mare, 1984).

 Given that our dependent variable is an ordinal variable, we think that it is more appropriate to follow a more conservative approach and use an ordinal regression model, such as the ordered logit. This is what we do for our main analysis described in the manuscript. However, we have also performed a multiple linear regression analysis as a robustness check for the results. In particular, in the subsection “Main results”, on page 22, we write: “In response to the request of one of the referees, as a robustness check, we estimate a multiple linear regression model with the same dependent and explanatory variables as in our baseline specification. The results from this model are largely in line with those obtained by estimating the baseline ordered logistic regression model as we find that trust in the government, science, and pharmaceutical companies are found to be positively associated with vaccination intentions, while the coefficients on trust in healthcare and the media are statistically insignificant (see S6 Table). The coefficient on trust in strangers now is found to be statistically insignificant at any conventional significance level.”

13. In the descriptive analysis, considering that the data collection is representative, I would elaborate a bit more on the distribution of vaccination willingness in Lithuania.

Thank you so much for this comment. We have now added additional information on the distribution of vaccination willingness. In subsection “Descriptive analysis” we describe how vaccination intent varies by age, gender, marital status and income. In particular, on page 14-15, we write:

 “Most of respondents expressed a willingness to get vaccinated against COVID-19 (see Fig 1). In total, around 69% of respondents said that they strongly agree, agree, or agree somewhat to receive a vaccine as soon as it becomes available. Almost 19% of respondents expressed negative attitudes toward COVID-19 vaccines, that is, they answered that they strongly disagree, disagree, or disagree somewhat to get vaccinated. Out of these with negative attitudes, more than a half (10.6% of all respondents) disagreed strongly with getting vaccinated. Around 12% of respondents said they neither agree, nor disagree to get vaccinated and thus could be considered as undecided. 

 The intent to get vaccinated varies with the socio-demographic characteristics of respondents. One of such characteristics is age. Older participants of the survey were less skeptical about the vaccine and were more willing to get vaccinated than younger participants. Among those older than 60 years, 83% expressed an intent to get vaccinated, that is, they answered that they strongly agree, agree or agree somewhat to get vaccinated once the vaccine becomes available. Among those aged 40-59 years this number stood at 66% and at 58% among those younger than 39 years.

 In terms of gender, men had somewhat stronger positive views toward getting the vaccine than women—64% of men said that they agree or agree strongly to get vaccinated, while for women this number stood at 62%. Also, more women than men had strong negative attitudes toward the vaccine—16.5% of women disagreed strongly or disagreed with getting vaccinated, while among men 14.5% did. Among those respondents who were married or lived with a partner, somewhat more (70.3%) strongly agreed, agreed or agreed somewhat to get vaccinated as compared to those who were single or divorced (66%). Also, vaccination intent varied with household income. 79.4% of respondents with after-tax household income above 3,000 euros strongly agreed, agreed or agreed somewhat to get vaccinated, while among those with income lower than 500 euros only 60.3% did.”

14. In the descriptive analysis, authors mention that “Both variables, representing trust in strangers, that is, trust strangers_d (rs = 0.073, p = 0.023) and trust strangers_i (rs = 0.103, p = 0.001) are not significantly correlated with vaccination intentions” while presenting p value that are under .05.

Thank you for this comment. In the “Descriptive analysis” subsection, on page 16, now we write: “Correlation between trust in strangers and vaccination intentions (rs = 0.073, p = 0.023) is rather weak, but still positive and statistically significant at 5% significance level.” All references to the indirect measure of trust have been removed from the text as we are keeping only the direct measure of trust in strangers in the analysis.

15. Several time in the manuscript, author mention p =0.000, which is statistically impossible, and indicate a very small p-value close but not equal to 0

Thank you for this comment. We have now changed it to p < 0.001 in the text and in the tables of the manuscript.

16. Authors indicate that the model with all control variables predicts vaccination intentions (DV) better than other models. However, adding a predictor in a model can only result in an improve in R-squared (a model with all control variables always has a greater Rsquared than a model with less of these control variables). Therefore, when comparing two models explaining the same DV, the question is whether the model with more predicting variables predicts significantly better than the other one. Is it the case here?

Thank you so much for this useful comment. We have now removed references to pseudo R-squared from the text. To compare the reduced form models with our baseline model, we conducted the likelihood ratio test. In subsection “Main results”, on page 22, we write: “To evaluate if the baseline model (see Table 3) fits the data better than the reduced form models with single trust variables and all controls (see Table 2), we conducted likelihood ratio tests. The results show that adding all trust variables as predictor variables to a model, results in a statistically significant improvement in the fit of the model (see S5 Table).”

17. I liked the conclusion section very much. I think that the reflections regarding trust and the crises in general (eg. Climate crisis) were very relevant.

Thank you very much for this positive comment.

18. I would disagree that the perceived side effects would not bias the results, due to the fact that the data collection occurred early in the vaccination campaign. Individuals might precisely be worried about side effects as very few people had been through, and documented the vaccine process yet.

Thank you so much for this comment. To address this concern, we have now corrected the “Conclusions” section. On page 26 we write: “First, in the regressions, we do not control for the individuals’ concerns about the safety, side-effects, and effectiveness of the vaccine against COVID-19, which are factors that have a significant effect on vaccination decisions. Thus, the estimates of the trust variables could partly capture the effects of these concerns. At the time of our survey there were only few personal experiences from the use of vaccines, thus some people could have had significant concerns about the safety of vaccines and their potential side effects.”

Thank you again for your very helpful comments. We hope we addressed them in a satisfactory way.

Response to Reviewer 2

Dear Reviewer,

 Thank you so much for reviewing our paper and for your thoughtful and constructive comments. Revising the paper based on your comments (in italics below) improved the paper significantly.

1. In the introduction you assert that social pressures probably played a small role when the survey was conducted, please provide more information or supporting research for this statement. I think in general adding more details on what the literature states on the relationship between social capital, trust, and health behaviors could be helpful if you have room. Some of this is in the discussion/conclusion and could be mentioned in the intro.

Thank you so much for this comment. We have now removed the statement about social pressures playing a small role when the survey was conducted from the manuscript. We thought that it did not add value to the analysis. Regarding the literature, we have now added more information on the link between social capital, trust, and health behaviors in our “Background and hypotheses” subsection on pages 5-8. In this subsection we cover the theoretical background, literature on the variables of interest and their links to vaccinations and other health behaviors as well as outline the hypotheses. We write:

 “Many authors have examined individuals’ voluntary vaccination decisions, which are often studied from the perspective of public goods theory [21-22]. If vaccinations can stop the spread of infections, then such a containment of a virus is a public good, which requires people’s cooperation in terms of them getting vaccinated. However, individuals personally have an incentive to free-ride and not incur the individual costs of vaccinations, such as safety concerns, potential side effects, costs of travel, and other monetary and non-monetary costs [23], while benefiting from the contained spread of the virus when a considerable number of people is vaccinated. In this way, free-riding can lead to a suboptimal collective outcome [24]. 

 The social capital theory suggests that social capital, that is, certain shared values, norms, bonds, and trust among people, can help societies overcome the free-rider problem and facilitate cooperation in the pursuit of socially valuable activities [5-6, 25-27]. When it comes to general trust in other people, that is, generalized trust, and, particularly, trust in strangers, more trusting individuals are more willing to cooperate and contribute to the public good, because they view other people as trustworthy and do not think they will be cheated [5]. In the context of vaccinations this could mean that if people trust others, they are more willing to get vaccinated, because they do not think that others will free-ride and refuse vaccinations needed to stop the spread of the virus.

 Different authors have shown social capital to be positively associated with health behavior, such as voluntary compliance to non-pharmaceutical interventions, including social distancing, during the COVID-19 pandemic [28-32]. Some empirical studies have also found a positive association between generalized trust and vaccination willingness [33-35], but some find no significant relationship between this type of trust and vaccination intentions [36]. A few previous studies have found a negative effect of generalized trust on social distancing behavior during the COVID-19 pandemic, arguing that when people trust others, they may believe that other people are sticking to restrictions and feel that it is safe to go out during the pandemic [37-38]. Hence, based on the social capital theory and previous empirical literature on trust and vaccinations, we hypothesize that:

 Hypothesis 1: Trust in strangers predicts higher vaccination intentions (H1).

 Findings from the literature have shown that vaccinations can also depend on trust in various institutions and systems that produce and deliver vaccines as well as decide on their need [7]. In particular, this includes trust in pharmaceutical companies and science that develop the vaccines and ensure their safety and efficacy, trust in the healthcare system that administers vaccinations, and trust in policy-makers, mostly the government, that decide on the needed vaccine and establish the legal and regulatory framework for vaccinations [7, 10]. 

 From the theoretical point of view, information asymmetry about the vaccine between individuals—less informed party—and institutions involved in the vaccination process—more informed party—makes trust in such institutions play an important role for the willingness to get vaccinated [7, 10]. Trust in institutions helps individuals with incomplete information weight the risks and benefits of vaccinations [39]. In this way, trust works as a heuristic shortcut to making a judgement by an individual with incomplete information about the safety, effectiveness, and importance of a vaccine in question [10, 40]. When individuals trust the institutions involved in the vaccination process, they believe that their representatives have the required competence and expertise, they have individuals’ best interests at heart and adhere to the principles of integrity [7, 41]. 

 Several empirical studies have confirmed the above theoretical consideration by providing evidence that there is a positive association between institutional trust and attitudes toward vaccination. For example, Jelnov and Jelnov [42] show that trust in the government leads to higher voluntary vaccination levels due to lower probability of a transparent and accountable government to promote an unsafe low-quality vaccine. Other authors have also found a positive association between trust in the government and willingness to get vaccinated [9, 33, 36, 41, 43-44]. Similarly, trust in science or scientists [10, 33, 36, 41, 45], in the healthcare system or its workers [34, 45-46], and in pharmaceutical companies [13-14] have been found to be positively related with vaccination intentions. Based on this literature, in our study, we raise the following hypotheses:

 Hypotheses 2-5: Trust in the government (H2)/ healthcare system (H3)/ science (H4)/ pharmaceutical companies (H5) predicts higher vaccination intentions.

 Meanwhile, the media is an important source of information about vaccination. The social learning theory suggests that trust in such information sources mediates the effect of exposure to information about a vaccine on attitudes toward vaccination [47]. That is, when the media provides people with information related to the vaccine, for example, outlines the benefits of vaccinations, and people believe the media to be a credible source of information, this information can have a positive effect on vaccine willingness. Different authors have also shown empirically that individuals’ attitudes toward vaccinations are positively related to trust in the media [8, 47-48]. Based on this research, we will test whether:

 Hypothesis 6: Trust in the media predicts higher vaccination intentions (H6).”

2. It is easier for the reader if you use a phrase to describe the variables in the text and tables instead of the variable name (for example "trust government" reads better as "trust in the government" and "trust stranger_i" reads better as "indirect measure of trust in strangers").

Thank you for this comment. We have now changed variable names to phrases in the text and in the tables. Our main explanatory variables are now called trust in strangers, trust in government, trust in healthcare, trust in pharma, trust in science and trust in media. To ensure that the variable names do not take too much space in the tables, we tried to keep them concise. Please note that we removed the measure of indirect trust in strangers as suggested by the Reviewer 1 (see answer to comment 10 of Reviewer 1). 

3. Please provide a new Table 1 where you provide information on the population (age, gender, race/ethnicity or nationality, responses to trust questions and all confounding variables) in the descriptive analysis section of the results. This is really important for the reader to better understand whose views are captured and many may not be aware of what the general demographics of Lithuania look like.

Thank you so much for this useful comment. We have compiled a table that summarizes the characteristics of the sample population, including distribution of survey participants by age, gender, nationality, responses to trust questions and all other questions used to construct other variables that were used in the analysis. Because of its large size, we provide the table as Supporting Information (S2 Table. Sample characteristics). In the “Descriptive analysis” subsection we have now included a paragraph describing the main sample characteristics and have provided a reference to the table. On page 13-14 we write: 

 “Twenty-seven respondents (out of 1,000) who answered the survey in less than 4 minutes were dropped from our analysis. This left us with the sample size of 973 observations. Around a half of these survey participants were women (55.4%) and almost half (48.2%) were aged 18-49. Lithuanians represented more than nine in ten (92.8%) of respondents. Seven in ten respondents had higher education (70.3%), were married or lived with a partner (70.9%), had household income lower than 2,000 euros (67.8%), and lived in a city or a town (67%). Almost one in three respondents lived in one of the three largest Lithuanian cities, that is, Vilnius (18%), Kaunas (9.5%), and Klaipeda (3.4%). Around one-fifth of participants had some personal experience with COVID-19, that is, they were either diagnosed with COVID-19 (7.3%) or they thought they have had COVID-19, but it has not been diagnosed (14.5%). S2 Table provides the additional characteristics of our data sample.”

4. Some of the tables are hard to read and could be reduced to improve reader understanding. For example in the current Table 1, all covariates are included in all models and are mentioned in the table note, so they don’t need to be in the table. Also this table has a lot of blank space, so this could be restructured so that each trust variable result is a column in a single row instead. Similarly current Table 2 seems to include the main model and other models to assess the robustness of your results. Maybe moving the tables for the additional analyses to the supplementary material could help readability, so that the reader can focus on your main model.

Thank you so much for this useful comment. The simple binary logistic model and the tables related to it were removed from the analysis as suggested by the Reviewer 1 (see answer to comment 9 of Reviewer 1). But we ensure that now other tables presenting the results, that is, Table 1, Table 2 and Table 3 (for baseline model), are easy to read and understand. 

5. The manuscript should also be reviewed to address errors in grammar. There are only a few of these, including one in the first sentence. It also needs to be edited for clarity in the results, when you’re talking about how the variables were classified in the analysis. Also please make sure you follow the journal’s guidelines for titling your sections!

Thank you so much for this comment. The manuscript has been reviewed to address the errors in grammar and edited for clarity. We have also ensured that the manuscript is in line with journal’s guidelines. The final version of the text will be also edited by a professional proofreader after we receive the decision from the journal.

6. Figures 1 and 2 use the 7-point Likert scale and then use the dichotomous version in the text of the results. Please be consistent in the presentation or remind the reader how they were categorized.

Thank you so much for this useful comment. We have now ensured that we remind the reader how the answers to trust and vaccination questions were categorized in the text. 

 To describe variations in responses to the vaccination question, in the “Descriptive analysis” subsection, on page 14-15, we write: “Most of respondents expressed a willingness to get vaccinated against COVID-19 (see Fig 1). In total, around 69% of respondents said that they strongly agree, agree, or agree somewhat to receive a vaccine as soon as it becomes available. Almost 19% of respondents expressed negative attitudes toward COVID-19 vaccines, that is, they answered that they strongly disagree, disagree, or disagree somewhat to get vaccinated. Out of these with negative attitudes, more than a half (10.6% of all respondents) disagreed strongly with getting vaccinated. Around 12% of respondents said they neither agree, nor disagree to get vaccinated and thus could be considered as undecided. 

 The intent to get vaccinated varies with the socio-demographic characteristics of respondents. One of such characteristics is age. Older participants of the survey were less skeptical about the vaccine and were more willing to get vaccinated than younger participants. Among those older than 60 years, 83% expressed an intent to get vaccinated, that is, they answered that they strongly agree, agree or agree somewhat to get vaccinated once the vaccine becomes available. Among those aged 40-59 years this number stood at 66% and at 58% among those younger than 39 years.

 In terms of gender, men had somewhat stronger positive views toward getting the vaccine than women—64% of men said that they agree or agree strongly to get vaccinated, while for women this number stood at 62%. Also, more women than men had strong negative attitudes toward the vaccine—16.5% of women disagreed strongly or disagreed with getting vaccinated, while among men 14.5% did. Among those respondents who were married or lived with a partner, somewhat more (70.3%) strongly agreed, agreed or agreed somewhat to get vaccinated as compared to those who were single or divorced (66%). Also, vaccination intent varied with household income. 79.4% of respondents with after-tax household income above 3,000 euros strongly agreed, agreed or agreed somewhat to get vaccinated, while among those with income lower than 500 euros only 60.3% did.” 

 To describe variations in responses to the vaccination question, on page 15 we write: “Regarding trust, respondents tended to trust institutions more than strangers. Around 23% expressed trust toward strangers, that is, answered that they trust completely, trust, or trust somewhat people they do not know personally (see Fig 2). For institutional trust, more than 47% of respondents answered that they trust government authorities, 48% trusted pharmaceutical companies, almost 55% trusted the healthcare system, and more than 84% of respondents trusted science. Only 38% of respondents expressed trust toward the media (see Fig 2). Here we consider that a person trusts an institution if he or she answered “trust completely,” “trust,” or “trust somewhat” to the trust questions in the survey. S3 Table provides the summary statistics of all variables used in the analysis.”

Thank you for your very helpful comments. We hope we addressed them in a satisfactory way. 

Additional references:

1. Jamieson, S. (2004). Likert scales: How to (ab) use them? Medical education, 38(12), 1217-1218.

2. Liddell, T. M., and Kruschke, J. K. (2018). Analyzing ordinal data with metric models: What could possibly go wrong? Journal of Experimental Social Psychology, 79, 328-348.

3. Long, J. S., and Freese, J. (2014). Regression models for categorical dependent variables using Stata. Stata press.

4. McKelvey, R. D., and Zavoina, W. (1975). A statistical model for the analysis of ordinal level dependent variables. Journal of mathematical sociology, 4(1), 103-120.

5. Norman, G. (2010). Likert scales, levels of measurement and the “laws” of statistics. Advances in health sciences education, 15(5), 625-632.

6. Winship, C., and Mare, R. D. (1984). Regression models with ordinal variables. American sociological review, 512-525.

---

## [Decision Letter · Decision Letter 1]

3 Oct 2022

PONE-D-22-13409R1Trust and vaccination intentions: Evidence from Lithuania during the COVID-19 pandemicPLOS ONE

Dear Dr. Kajackaite,

Thank you for submitting your manuscript to PLOS ONE. After careful consideration, we feel that it has merit but does not fully meet PLOS ONE’s publication criteria as it currently stands. Therefore, we invite you to submit a revised version of the manuscript that addresses the points raised during the review process.

The revised submission improved in a very proper manner. However, further revisions are still necessary. In this regard, the author should focus on discussion and the formulated statements.

We look forward to receiving your revised manuscript.

Kind regards,

Stefan Cristian Gherghina, PhD. Habil.

Academic Editor

PLOS ONE

Journal Requirements:

Reviewers' comments:

Reviewer's Responses to Questions

**Comments to the Author**

1. If the authors have adequately addressed your comments raised in a previous round of review and you feel that this manuscript is now acceptable for publication, you may indicate that here to bypass the “Comments to the Author” section, enter your conflict of interest statement in the “Confidential to Editor” section, and submit your "Accept" recommendation.

Reviewer #3: All comments have been addressed

Reviewer #4: (No Response)

2. Is the manuscript technically sound, and do the data support the conclusions?

Reviewer #3: Yes

Reviewer #4: Partly

3. Has the statistical analysis been performed appropriately and rigorously? 

Reviewer #3: Yes

Reviewer #4: Yes

4. Have the authors made all data underlying the findings in their manuscript fully available?

Reviewer #3: Yes

Reviewer #4: Yes

5. Is the manuscript presented in an intelligible fashion and written in standard English?

Reviewer #3: Yes

Reviewer #4: Yes

6. Review Comments to the Author

Reviewer #3: This piece reviews the effect of several different forms of trust (both institutional and interpersonal) on COVID-19 vaccine behavioral intentions in Lithuania. In a cross-sectional online survey, the authors find that both institutional and interpersonal trust are associated with increased vaccine uptake, even when accounting for a diverse range of socio-political and demographic controls in multivariate models.

In general, I found this piece to be both well written and well motivated. Recognizing that this piece already received a revise decision at this journal, I believe that my role as a reviewer should be to (a) make an effort to assess how well the authors' revised manuscript addressed the concerns raised previously in earlier rounds of review, and (b) to offer any additional (and, likely, more-minor) suggestions regarding how the authors might improve this paper.

Regarding the former, I think that the authors have done an excellent job responding to important points raised in earlier rounds of review by R1 and R2; many of which I shared, when contrasting the revised from original manuscript. I especially appreciated the authors' willingness to provide additional conceptual detail regarding the theoretical motivations underlying their expectations, the streamlining of data presented throughout the piece, and efforts to improve the piece's grammar and structural cohesion.

Correspondingly, I believe that this piece should be published -- with some minor revisions -- in this journal.

With that being said, I wanted to offer a few (more minor) suggestions regarding how the authors might continue to improve this piece.

Ordinarily, I am reluctant to offer additional feedback at this stage of the review process. However, because I was brought on as an additional reviewer, I wanted to raise a few small areas for improvement that I encountered while reading this manuscript. So long as the authors are able to address many or most of these concerns, I would be happy to recommend publication in this journal.

Minor concerns:

--line 125: I'm still having a difficult time following the theoretical motivation underlying Hypothesis 1. The authors outline two competing effects regarding how what they call "generalized trust" (more on this in a moment) might be associated with increased vaccine uptake. Notably, they suggest that higher levels of trust could be associated with inaccurate perceptions of pandemic health risks, thereby leading people to engage in *more* risky health behaviors (lines 121-123); including, presumably, not vaccinating.

Yet, in line 124, the authors write: "Hence, based on the social capital theory and previous empirical literature..." This link isn't entirely clear to me. How do we move, conceptually, from a pattern of conflicting results to the sentence quoted above? I would encourage the authors to make additional effort to explicate this conceptual link.

Moreover, and I apologize if this is a minor semantic quibble, but I object slightly to the use of the term "generalized trust" -- which is typically used to refer to trust across many different institutional and interpersonal actors (e.g., see: https://onlinelibrary.wiley.com/doi/full/10.1111/ajps.12234) -- as a synonym for trust in people one does not know ("strangers"). I'd encourage the authors to use consistent terminology throughout the piece, perhaps eschewing references to "generalized trust" in this domain.

-- line 165: The authors claim that this survey is nationally representative of the Lithuanian adult population. While I appreciated the authors' willingness to offer a new table in the Supplementary Materials denoting the basic descriptive properties of their sample, it was not clear to me how these demographics compared to those of the Lithuanian population. I would encourage the authors to therefore amend this supplemental table by including a comparison of sample properties to nationally representative benchmarks.

Relatedly, I think the authors could say much more about the process by which the third party survey firm with whom they contracted not only invited individuals to participate in the study (which the authors addressed -- quite well, in my view! -- in the previous round of review) but how these individuals were *sampled.* The authors later note that the firm was taking random draws from an online opt in panel service (line 197).

In order for the study to be nationally representative, this would imply that the online opt in sampling frame *itself* is nationally representative; something which is very atypical in online survey research (although I confess that I am not familiar with how this particular firm operates!). Correspondingly, I'd encourage the authors to provide additional information about how their sampling protocols ensure national representativeness.

-- Tables 1-3: While I appreciate the authors' willingness to calculate and present more-substantively tractable quantities from their ordered logistic regression models, the piece provides little information (at least, in my reading) of whether or not each independent variable in the model had a statistically significant effect on vaccine uptake. I raise this point because, in theory, statistically insignificant independent variables (e.g., one of the dimensions of trust) could potentially have non-linear and significant marginal effects at some points on the ordered Likert scale on vaccination intentions, while nevertheless not resulting in appreciable change in vaccination intentions that are appreciable from zero. For that reason, some scholars caution against calculating marginal effects and predicted probabilities from statistically insignificant regressors.

I would therefore encourage the authors to add information (either to the manuscript's main text, or to the Tables) denoting whether or not each of the measures of trust yielded a statistically significant increase in vaccination intentions in the ordered logistic regression models, in addition to presenting the marginal effects listed in each table. Alternatively, the authors might consider migrating Table S6 from the supplement to the main text. I completely defer to the authors on this score, regarding which course of action might be best. As long as they make some effort to introduce this information in the main text, I'll be satisfied!

Reviewer #4: Thank you for this enlightening study. I was recruited as a new reviewer for the revised version. I have reviewed the original reviewer comments and the authors' revisions. I believe the authors have done a nice job and been responsive to the original reviewers' suggestions.

I noticed a few things as part of my own reading of the manuscript that I think will strengthen it and some of its arguments before publication.

1. It seems a bit contradictory to frame the "types" of trusted institutions that correlate with vaccination intention as ones "that are directly related to the development, approval, and distribution of vaccines" (abstract). I mean this is true, but healthcare as an institution certainly falls under this umbrella category as well. How do we reconcile this?

2. There seems to be another contradiction when motivating the case context vis-a-vis the descriptive results. On page 4, the manuscript argues that Lithuania offers an interesting case in part because "Lithuania has suffered greatly from the COVID-19 pandemic and has faced high vaccine skepticism (see subsection “COVID-19 and vaccinations in Lithuania”)". At the same time, the authors note "Most of respondents expressed a willingness to get vaccinated against COVID-19 (see Fig 1). In total, around 69% of respondents said that they strongly agree, agree, or agree somewhat to receive a vaccine as soon as it becomes available. How do we reconcile that vaccination skepticism is high yet at the same time most people say they want the vaccine? Is it because of a temporal dynamic from time of study to actual vaccinations? Is it because the panel is not really representative of the population (i.e., vaccine-willing individuals are more likely to self-select into panel)?. Or is it because initial vaccine willingness is prone to a social desirability factor?

3. The latter option takes me to the biggest concern with the study results. How do we know these results are confounded by a latent social desirability factor? People who likes to conform to social norms might indicate that they are more trusting (because it is considered the right thing to be) and express greater willingness to get vaccinated (because it is touted as the socially and morally right thing to do)? In other words might social desirability bias be an omitted variable bias in this study? One way to potentially address this somewhat is to draw on studies that have examined the extent to which Covid-19 mitigation behaviors are susceptible to social desirability (e.g., Jensen 2020 and Larsen et al. 2020)

4. Finally it can offer additional credence to validity of the self-reported measure of vaccine willingness to demonstrate how it maps onto actual vaccination behavior. Several recent studies show that early willingness strongly predict actual uptake, see for example Jensen, Ayers and Koskan (2022) in PLoS ONE.

References:

Larsen, M., Nyrup, J., & Petersen, M. B. (2020). Do survey estimates of the public’s compliance with COVID-19 regulations suffer from social desirability bias?. Journal of Behavioral Public Administration, 3(2).

Jensen, U. T. (2020). Is self-reported social distancing susceptible to social desirability bias? Using the crosswise model to elicit sensitive behaviors. Journal of Behavioral Public Administration, 3(2).

Jensen, U. T., Ayers, S., & Koskan, A. M. (2022). Video-based messages to reduce COVID-19 vaccine hesitancy and nudge vaccination intentions. PloS one, 17(4), e0265736.

7. PLOS authors have the option to publish the peer review history of their article (what does this mean?). If published, this will include your full peer review and any attached files.

Reviewer #3: **Yes: **Matt Motta

Reviewer #4: No

---

## [Author Response · Author response to Decision Letter 1]

4 Nov 2022

Response to the Academic Editor

Dear Professor Gherghina ,

 Thank you again for giving us the opportunity to revise our paper for a possible publication in PLOS ONE. 

 In the following, we document how we revised the paper in reaction to each of your and reviewers’ comments. Original comments by the review team are written in italics. 

Thank you for this comment. We have now reviewed our reference list. We found some errors in it which have now been addressed. Some additional references have been added to the list at the request of the reviewers or they were required by the changes introduced in the text of the manuscript. Here is a detailed list of the adjustments we have made to the reference list:

• We have replaced the reference to Galvani et al. (2007) with a reference to Bauch et al. (2003) (see [24] in the manuscript), because we think the latter is more relevant for the statement we are making in the text of the manuscript (see line 107 in the manuscript).

• Bartscher et al. (2021) has been moved down in the reference list (see [32] in the manuscript) due to changes in the text (see lines 121-122 in the manuscript).

• We have added a reference to Šiđanin et al. (2021) (see [47] in the manuscript) to illustrate the relationship between trust in the media and individuals’ attitudes toward vaccinations established in the empirical literature (see lines 165-167 in the manuscript). 

• We have adjusted the reference to Saka et al. (2022) to reflect its “accepted for publication” status (see [64] in the manuscript).

• We have removed the reference to Guiso et al. (2008) due to changes in the text (see lines 585-587 in the manuscript). 

• We have introduced new references to the reference list (see [66-72] in the manuscript) due to new text added to the manuscript (see line 608-626 in the manuscript). Some of these references (see [66-67, 71] in the manuscript) were recommended by Reviewer 4 (see comments 3 and 4 below).

Thank you for your very helpful comments. We hope we addressed them in a satisfactory way. 

  

Response to Reviewer 3

Dear Professor Motta,

 Thank you so much for reviewing our paper and for your thoughtful and constructive comments. Revising the paper based on your comments (in italics below) improved the paper significantly.

1. Line 125: I’m still having a difficult time following the theoretical motivation underlying Hypothesis 1. The authors outline two competing effects regarding how what they call "generalized trust" (more on this in a moment) might be associated with increased vaccine uptake. Notably, they suggest that higher levels of trust could be associated with inaccurate perceptions of pandemic health risks, thereby leading people to engage in *more* risky health behaviors (lines 121-123); including, presumably, not vaccinating. Yet, in line 124, the authors write: "Hence, based on the social capital theory and previous empirical literature..." This link isn’t entirely clear to me. How do we move, conceptually, from a pattern of conflicting results to the sentence quoted above? I would encourage the authors to make additional effort to explicate this conceptual link.

Moreover, and I apologize if this is a minor semantic quibble, but I object slightly to the use of the term "generalized trust" -- which is typically used to refer to trust across many different institutional and interpersonal actors (e.g., see: https://onlinelibrary.wiley.com/doi/full/10.1111/ajps.12234) -- as a synonym for trust in people one does not know ("strangers"). I’d encourage the authors to use consistent terminology throughout the piece, perhaps eschewing references to "generalized trust" in this domain.

Thank you so much for this useful comment. We agree that the conceptual link between generalized trust and vaccinations lacked clarity. We have now explained it in the manuscript. In particular, on pages 6-7 of the manuscript we write the following: 

 “The social capital theory suggests that social capital, that is, certain shared values, norms, bonds, and trust among people, can help societies overcome the free-rider problem and facilitate cooperation in the pursuit of socially valuable activities [5-6, 25-27]. When it comes to general trust in other people, that is, generalized trust, and, particularly, trust in strangers, more trusting individuals are more willing to cooperate and contribute to the public good, because they view other people as trustworthy and do not think they will be cheated [5]. In the context of vaccinations, this could mean that if people trust others, they are more willing to get vaccinated, because they do not think that others will free-ride and refuse vaccinations needed to stop the spread of the virus.

 But empirical evidence on the role of social capital in shaping health behavior is somewhat conflicting. Some authors show that social capital (where trust is an important component) is associated with increased voluntary compliance to non-pharmaceutical interventions, including social distancing during the COVID-19 pandemic [28-31], as well as improved health outcomes, that is, fewer COVID-19 cases and fewer excess deaths per capita [32]. Some empirical studies also find a positive association between generalized trust and vaccination willingness [33-35]. However, some studies on health behavior find the opposite and do not confirm the theoretical considerations of the social capital theory. For example, Jennings et al. [36] show evidence of no significant relationship between generalized trust and vaccination intentions during the COVID-19 pandemic. Deopa and Fortunato [37] and Doganoglu and Ozdenoren [38] find a negative effect of generalized trust on social distancing behavior during the COVID-19 pandemic, arguing that when people trust others, they may believe that other people are sticking to restrictions and feel that it is safe to go out. However, despite this mixed empirical evidence, we base our prediction on the social capital theory. Hence, we hypothesize that:

 Hypothesis 1: Trust in strangers predicts higher vaccination intentions (H1).”

 Also, thank you for your comment on generalized trust. Many authors define generalized trust as trust in other people in general (Uslaner, 2002; Guiso et al., 2011; Carl and Billari, 2014; Delhey et al., 2011). Generalized trust is most often measured using the question “Generally speaking would you say that most people can be trusted or that you can’t be too careful in dealing with people? (Rosemberg, 1956).” Many scholars also view trust in people one does not know, that is, trust in strangers, as the cornerstone of generalized trust, and argue that generally trusting means that we trust strangers (Uslaner, 2002). Thus, we think that the terms “generalized trust” and “trust in strangers” can be used as synonyms in the manuscript, because they are very closely related concepts, or some would even argue that “trust in strangers” can be referred to as “generalized trust”, at least in a narrow sense. To the best of our knowledge, generalized trust rarely includes trust across different institutional actors. It also distinguishes from particularized trust, that is trust toward familiar others, including friends, neighbors, and co-workers. Since references to “generalized trust” are essential for our review of the related literature, including the theoretical background, and the use of such references are in line with the existing literature, we are inclined to keep them in the manuscript. 

 However, to ensure that our reasoning is clear, we introduce some minor changes to the text of the manuscript. In the subsection “Present research” on page 9 of the revised manuscript we write: “For interpersonal trust, we explore trust in strangers, because it should help societies overcome the free-riding problem and encourage cooperation between strangers [5-6]. In this paper we focus on trust in strangers instead of the broader definition of generalized trust. The broader definition of generalized trust measures trust in other people in general, which is most often elicited by asking “Generally speaking would you say that most people can be trusted or that you can’t be too careful in dealing with people? [49]”. Generalized trust and trust in strangers are very closely related concepts and are often used as synonyms [5]. However, we think that trust in strangers eliminates the ambiguity inherent in the broader concept of generalized trust.”

2. Line 165: The authors claim that this survey is nationally representative of the Lithuanian adult population. While I appreciated the authors’ willingness to offer a new table in the Supplementary Materials denoting the basic descriptive properties of their sample, it was not clear to me how these demographics compared to those of the Lithuanian population. I would encourage the authors to therefore amend this supplemental table by including a comparison of sample properties to nationally representative benchmarks. Relatedly, I think the authors could say much more about the process by which the third party survey firm with whom they contracted not only invited individuals to participate in the study (which the authors addressed -- quite well, in my view! -- in the previous round of review) but how these individuals were *sampled.* The authors later note that the firm was taking random draws from an online opt in panel service (line 197). In order for the study to be nationally representative, this would imply that the online opt in sampling frame *itself* is nationally representative; something which is very atypical in online survey research (although I confess that I am not familiar with how this particular firm operates!). Correspondingly, I’d encourage the authors to provide additional information about how their sampling protocols ensure national representativeness.

Thank you so much for this comment. We have now included a comparison of sample properties to nationally representative benchmarks for main demographic characteristics in the S2 Table. We have also provided some additional information on the sampling process that was made available to us by the company “Norstat” that implemented the survey. In subsection “Survey” on pages 9-10 of the revised manuscript we write the following: 

 “We employed a data set from a representative incentivized online panel survey conducted on 13–20 January 2021. We hired the company “Norstat” to implement the survey using its online access panel, that is, a group of registered internet users who have agreed to take part in various surveys. For participating in surveys, “Norstat” panel members are rewarded with virtual coins that could be exchanged into gift cards, coupons, or donated to a charity. The database of “Norstat” panel members was collected by the company by conducting member recruitment campaigns and representative surveys of the general population. The company sends individual invitations to potential panel members asking them to join the panel and individuals can then either accept or reject the invitations. The invitation-based system allows the company to ensure a diverse pool of individuals available for nationally representative surveys. Our survey participants were selected from the panel randomly according to the representativeness parameters, including age groups, gender, districts, and size of settlement (urban or rural). Invitations to participate in our survey were sent to potential participants by an automated system via email, which included a link to our questionnaire. The process of sending invitations continued until all sampling quotas for the target groups were fulfilled. The sampling quotas were set according to the population distribution data provided by Statistics Lithuania.”

3. Tables 1-3: While I appreciate the authors’ willingness to calculate and present more-substantively tractable quantities from their ordered logistic regression models, the piece provides little information (at least, in my reading) of whether or not each independent variable in the model had a statistically significant effect on vaccine uptake. I raise this point because, in theory, statistically insignificant independent variables (e.g., one of the dimensions of trust) could potentially have non-linear and significant marginal effects at some points on the ordered Likert scale on vaccination intentions, while nevertheless not resulting in appreciable change in vaccination intentions that are appreciable from zero. For that reason, some scholars caution against calculating marginal effects and predicted probabilities from statistically insignificant regressors. I would therefore encourage the authors to add information (either to the manuscript’s main text, or to the Tables) denoting whether or not each of the measures of trust yielded a statistically significant increase in vaccination intentions in the ordered logistic regression models, in addition to presenting the marginal effects listed in each table. Alternatively, the authors might consider migrating Table S6 from the supplement to the main text. I completely defer to the authors on this score, regarding which course of action might be best. As long as they make some effort to introduce this information in the main text, I’ll be satisfied!

Thank you for this useful comment. We have now added several tables with additional results from our estimated regression model as Supporting Information (see S5-S7 Table). These tables report the estimated logit coefficients of trust variables obtained by estimating the baseline model (see S7 Table) as well as the estimated logit coefficients from the reduced form ordered logistic regression models (see S5 and S6 Table). In the subsection “Main results” we have now provided information on the logit coefficients of the trust variables as well as the references to S5-S7 Table. In particular, on pages 17-23 of the revised manuscript we write the following:

 “First, we present the results from the simplest regression specification, which regresses vaccination on every trust variable in separate regressions without the control variables. As expected, we find the positive association between all trust variables and vaccination intentions. The logit coefficients are statistically significant at least at a 5% significance level (see S5 Table). The average marginal effects of all trust variables are also statistically significant at least at a 5% significance level for all categories of responses to the vaccination question (see columns 1.1-1.6 in Table 1). The average marginal effects of institutional trust variables are larger in numerical terms than those of trust in strangers. Overall, the average marginal effects of trust in strangers are relatively small in numerical terms. On average, an increase in trust in strangers by 1 point is associated with a 2.2%-point greater probability of “agreeing strongly” to getting vaccinated. For trust in the government, the healthcare system, the pharmaceutical companies and the media, an increase in trust by 1 point is associated with around a 10-12%-point greater probability of reporting the highest vaccination intentions. Trust in science demonstrates the largest average marginal effects observed for the highest vaccination intentions (see column 1.4 in Table 1).

 Next, we present the results from the regression models that regress vaccination intentions on every of the six trust variables separately and all control variables (sociodemographic characteristics of respondents, their health, conspiracy beliefs, fears of getting sick with COVID-19, impact on finances in the case of COVID-19, and risk preferences). We find that when we control for potential covariates, the logit coefficient of trust in strangers becomes statistically insignificant at any conventional significance level (see S6 Table). This is also reflected in the average marginal effects of trust in strangers, which are also statistically insignificant for all categories of responses to the vaccination question (see column 2.1 in Table 2). But the logit coefficients of the remaining trust variables, that is, trust in the government, trust in healthcare, trust in science, trust in pharma, and trust in the media, remain statistically significant at least at the 1% significance level (see S6 Table). The average marginal effects of these trust variables also keep their signs and remain statistically significant at the 1% level for all categories of vaccination variables (see columns 2.2-2.6 in Table 2). The association between vaccination intentions and the institutional trust variables remains positive. On average, an increase in trust in different institutions involved in the vaccination process by 1 point is associated with a 7.4-9.4%-point greater probability of agreeing strongly to getting vaccinated. Regarding the media, as trust in it goes up by 1 point, the probability of reporting highest vaccination intentions increases by 5.6%-points.

 However, the above-reported results from the first two regression models that include each trust variable separately could be misleading. That is, the estimated marginal effects of the trust variables may be biased upward, as the individually included trust variables might be capturing the effects of other trust variables that are omitted from the model. For this reason, the third regression specification, which includes all trust variables and controls, is estimated. This is our baseline model. 

 Estimating the baseline model yields logit coefficients that are statistically significant at the 1% significance level for trust in the government, trust in science, and trust in pharmaceutical companies (see S7 Table). The average marginal effects of these trust variables also remain statistically significant at the 1% level for all categories of vaccination variables. As trust in these institutions rises, the probability that the individuals report higher vaccination intentions increases (see Table 3). On average, an increase in trust in the government, science, and pharmaceutical companies by 1 point is associated with a 3.7, 3.9, and 4.7%-point greater probability of reporting the highest vaccination intentions, that is, “agreeing strongly” to getting vaccinated, respectively. In other words, higher trust in these institutions reduces the probability of disagreeing, being undecided, and agreeing less than strongly to getting vaccinated. These findings are in line with previous results in the literature [9-10, 33, 43] and provide evidence in favor of hypotheses H2, H4, and H5.

 Although some authors found that trust in healthcare [34, 45] and trust in the media [8, 46-48] were associated with higher vaccination intentions, in our case, these effects are potentially reduced by the inclusion of other trust variables, in particular, trust in the government and trust in science. The estimates of our baseline model show that the logit coefficients of trust in healthcare (H3) and trust in the media (H6) are statistically insignificant at any conventional level (see S7 Table). The average marginal effects of these trust variables are also statistically insignificant for all categories of responses to the vaccination question (see Table 3).

 Furthermore, we find that when we estimate the baseline model, the logit coefficient of trust in strangers becomes statistically significant at the 5% significance level (see S7 Table). We also find that higher trust in strangers is associated with a lower probability of having high vaccination intentions (see the second column in Table 3). Thus, we do not find evidence in favor of H1. This result contrasts with the findings of some authors [33-35], who show that generalized trust is positively associated with the willingness to get vaccinated. One of the potential explanations of our result could be related to the fact that in this paper we focus on trust in strangers, which is a somewhat different concept than the broader concept of generalized trust analyzed in most other similar studies. Differences in the timing of surveys could also play a role—our survey was conducted quite early in the vaccination process, possibly before most people had internalized the social benefits of vaccinations. Another potential explanation for the negative association between trust in strangers and vaccinations is that when people trust others, they may believe that other people will protect against the disease, for example, by adhering to specialists’ recommendations about safe health behavior during the pandemic and/ or by getting vaccinated, thus, they may feel safer about not rushing to get their vaccine. A few previous studies analyzing the role of generalized trust in explaining social distancing behavior during the COVID-19 pandemic have also found similar results [37-38]. However, our estimated average marginal effects of trust in strangers are relatively small in numerical terms and are statistically significant at the 5% significance level only for the highest and lowest vaccination intentions (see the second column in Table 3), meaning that one should be cautious in drawing strong conclusions from this finding.”

 In addition, in S7 Table we have included the logit coefficients of the control variables obtained by estimating our baseline model. We have also included a reference to this table in the manuscript. In subsection “Additional results” on page 24 of the manuscript we write: “In this subsection, we report and discuss the additional findings obtained by estimating the baseline ordered logistic regression model. The control variables that we include in this model provide interesting insights about how individual characteristics, beliefs, and attitudes are associated with vaccination intentions. In S7 Table we report the logit coefficients of all control variables that are included in our baseline model. In S1 Fig we plot the average marginal effects of those control variables that have statistically significant (at least at a 5% significance level) logit coefficients.”

 Also, to ensure that we describe all findings obtained by estimating our baseline model that are reported in S7 Table, we have oulined a few additional observations about the additional results to the text of the revised manuscript. In particular, in subsection “Additional results” on pages 24-25 we write: “In addition, individuals who report having higher income as well as those who prefer not to answer the question about their income are more likely to have high vaccination intentions. Some authors have also found a positive association between income and vaccinations [9, 57]. We also find that individuals from Klaipeda—the third largest Lithuanian city and a major seaport with a relatively large Russian-speaking population—are less likely to be in favor of getting vaccinated.” S1 Fig was also complemented with additional graphs plotting the average marginal effects for variables 2000-2999 euros, Prefer not to answer, and Klaipeda city.

Thank you again for your very helpful comments. We hope we addressed them in a satisfactory way. And thank you so much for believing that this paper should be accepted after minor revisions. Your comments have encouraged us. 

 

Response to Reviewer 4

Dear Reviewer,

 Thank you so much for reviewing our paper and for your thoughtful and constructive comments. Revising the paper based on your comments (in italics below) improved the paper significantly.

1. It seems a bit contradictory to frame the "types" of trusted institutions that correlate with vaccination intention as ones "that are directly related to the development, approval, and distribution of vaccines" (abstract). I mean this is true, but healthcare as an institution certainly falls under this umbrella category as well. How do we reconcile this?

Thank you so much for this comment. We have now removed references to the category of institutions that are “directly related to the development, approval, and distribution of vaccines.” In particular, in the abstract on page 2 of the revised manuscript we write: “Using unique survey data from Lithuania during the COVID-19 pandemic, we show that trust in government authorities, science, and pharmaceutical companies are important predictors of individual vaccination intentions.”. In the “Conclusions” section we write: “Our survey data show that the intent to get vaccinated is positively associated with trust in the government, science, and pharmaceutical companies.”

2. There seems to be another contradiction when motivating the case context vis-a-vis the descriptive results. On page 4, the manuscript argues that Lithuania offers an interesting case in part because "Lithuania has suffered greatly from the COVID-19 pandemic and has faced high vaccine skepticism (see subsection “COVID-19 and vaccinations in Lithuania". At the same time, the authors note "Most of respondents expressed a willingness to get vaccinated against COVID-19 (see Fig 1). In total, around 69% of respondents said that they strongly agree, agree, or agree somewhat to receive a vaccine as soon as it becomes available. How do we reconcile that vaccination skepticism is high yet at the same time most people say they want the vaccine? Is it because of a temporal dynamic from time of study to actual vaccinations? Is it because the panel is not really representative of the population (i.e., vaccine-willing individuals are more likely to self-select into panel)? Or is it because initial vaccine willingness is prone to a social desirability factor?

Thank you for this valuable comment. We agree that our writing in motivating the case context was not entirely accurate. When we write about high vaccine skepticism in the “Novelty of the study” subsection we mean vaccine skepticism in over-50-year-olds and slow vaccination uptake, and when we describe our survey data we talk about vaccine willingness in over-18-year-olds. We are sorry for our imprecision and are grateful for you noting it. We have now adjusted the text of the manuscript. In subsection “Novelty of the study” on page 4, we now write: “Lithuania has suffered greatly from the COVID-19 pandemic and it has faced sluggish vaccinations and high vaccine skepticism among the older population (see subsection “COVID-19 and vaccinations in Lithuania”).”

 Also, as you correctly noted, there could be a gap between early vaccination intentions, that we analyze in our paper, and actual vaccinations due to temporal dynamics and/ or social desirability bias. However, there is no way to check for this in our analysis, since individual data for vaccine uptake is not publicly available. But we outline this concern as one of the limitations of our study in the “Conclusions” section. In particular, on pages 27-28 we write: “However, there could be other factors than the social desirability bias that could lead to differences between vaccination intentions and the actual vaccine uptake, such as the temporal dynamics. Although in our study, many respondents expressed willingness to get vaccinated, which, according to Jensen et al. [71], should predict a high uptake of COVID-19 vaccines, the observed vaccination process was relatively slow in Lithuania. It may be the case that some individuals had second thoughts about getting vaccinated when the time to receive their vaccine came. To address the social desirability bias as well as other concerns related to the potential gap between vaccine willingness and actual vaccine uptake, future studies could use experimental measures of trust [70, 72] and actual vaccination behavior.”

3. The latter option takes me to the biggest concern with the study results. How do we know these results are confounded by a latent social desirability factor? People who like to conform to social norms might indicate that they are more trusting (because it is considered the right thing to be) and express greater willingness to get vaccinated (because it is touted as the socially and morally right thing to do)? In other words might social desirability bias be an omitted variable bias in this study? One way to potentially address this somewhat is to draw on studies that have examined the extent to which Covid-19 mitigation behaviors are susceptible to social desirability (e.g., Jensen 2020 and Larsen et al. 2020).

Thank you so much for this useful comment and the references. We agree that social desirability factor could bias the results. To ensure that we emphasize this limitation enough we have added additional information to the “Conclusions” section on pages 27-28 of the manuscript. We write:

 “Second, the study investigates self-reported trust and vaccination intentions, which may suffer from the social desirability bias and be prone to other concerns. The social desirability bias implies that survey respondents may overreport socially desirable and/ or underreport socially undesirable behavior. In our case, to be viewed favorably by others, respondents may indicate that they are more trusting and/ or more willing to get vaccinated than they actually are. But it is likely that this bias is not a very big concern here, as several studies have shown that the social desirability bias for self-reported health behaviors during the COVID pandemic, such as social distancing, is either very small [66] or even non-existent [67]. In addition, some studies have found that self-reported interpersonal and institutional trust predicts experimental trust measures [68-70]. However, there could be other factors than the social desirability bias that could lead to differences between vaccination intentions and the actual vaccine uptake, such as the temporal dynamics. Although in our study, many respondents expressed willingness to get vaccinated, which, according to Jensen et al. [71], should predict a high uptake of COVID-19 vaccines, the observed vaccination process was relatively slow in Lithuania. It may be the case that some individuals had second thoughts about getting vaccinated when the time to receive their vaccine came. To address the social desirability bias as well as other concerns related to the potential gap between vaccine willingness and actual vaccine uptake, future studies could use experimental measures of trust [70, 72] and actual vaccination behavior.”

4. Finally it can offer additional credence to validity of the self-reported measure of vaccine willingness to demonstrate how it maps onto actual vaccination behavior. Several recent studies show that early willingness strongly predict actual uptake, see for example Jensen, Ayers and Koskan (2022) in PLoS ONE.

Thank you so much for this comment and the suggested reference. On page 28 of the manuscript we have now added: “Although in our study, many respondents expressed willingness to get vaccinated, which, according to Jensen et al. [71], should predict a high uptake of COVID-19 vaccines, the observed vaccination process was relatively slow in Lithuania.”

Thank you for your very helpful comments. We hope we addressed them in a satisfactory way.  

Additional references:

1. Carl, N. and Billari, F.C., 2014. Generalized trust and intelligence in the United States. PloS one, 9(3), p.e91786.

2. Delhey, J., Newton, K. and Welzel, C., 2011. How general is trust in “most people”? Solving the radius of trust problem. American Sociological Review, 76(5), pp.786-807.

3. Guiso, L., Sapienza, P. and Zingales, L., 2011. Civic capital as the missing link. Handbook of social economics, 1, pp.417-480.

4. Rosenberg, M., 1956. Misanthropy and political ideology. American sociological review, 21(6), pp.690-695. 

5. Uslaner, E. M., 2002. The moral foundations of trust. Cambridge University Press.

---

## [Decision Letter · Decision Letter 2]

9 Nov 2022

Trust and vaccination intentions: Evidence from Lithuania during the COVID-19 pandemic

PONE-D-22-13409R2

Dear Dr. Kajackaite,

We’re pleased to inform you that your manuscript has been judged scientifically suitable for publication and will be formally accepted for publication once it meets all outstanding technical requirements. In this regard, the remaining suggestions of the third referee should be implemented.

Kind regards,

Stefan Cristian Gherghina, PhD. Habil.

Academic Editor

PLOS ONE

Additional Editor Comments (optional):

Reviewers' comments:

Reviewer's Responses to Questions

**Comments to the Author**

1. If the authors have adequately addressed your comments raised in a previous round of review and you feel that this manuscript is now acceptable for publication, you may indicate that here to bypass the “Comments to the Author” section, enter your conflict of interest statement in the “Confidential to Editor” section, and submit your "Accept" recommendation.

Reviewer #3: All comments have been addressed

Reviewer #4: All comments have been addressed

2. Is the manuscript technically sound, and do the data support the conclusions?

Reviewer #3: Yes

Reviewer #4: Yes

3. Has the statistical analysis been performed appropriately and rigorously? 

Reviewer #3: Yes

Reviewer #4: Yes

4. Have the authors made all data underlying the findings in their manuscript fully available?

Reviewer #3: Yes

Reviewer #4: Yes

5. Is the manuscript presented in an intelligible fashion and written in standard English?

Reviewer #3: Yes

Reviewer #4: Yes

6. Review Comments to the Author

Reviewer #3: I thank the authors for replying to my comments and concerns regarding the revised manuscript. I found the second revision to be significantly improved, and I now recommend publication.

One very-minor point: I think that the rollout of H1 reads much more clearly now. The authors outline why it is that they expect to observe an effect, but note that past research produces conflicting result. Still, some type of transition sentence preceding the listing of the hypothesis itself (e.g., "Despite this conflicting pattern of results, we nevertheless expect...") might help the piece read a bit more smoothly.

Reviewer #4: Thank you for the careful revisions. I'm recommending accept. Congratulations on your fine work and I look forward to seeing it online.

7. PLOS authors have the option to publish the peer review history of their article (what does this mean?). If published, this will include your full peer review and any attached files.

Reviewer #3: **Yes: **Matt Motta

Reviewer #4: No

---

## [Editor Report · Acceptance letter]

14 Nov 2022

PONE-D-22-13409R2 

Trust and vaccination intentions: Evidence from Lithuania during the COVID-19 pandemic 

Dear Dr. Kajackaite:

I'm pleased to inform you that your manuscript has been deemed suitable for publication in PLOS ONE. Congratulations! Your manuscript is now with our production department. 

Kind regards, 

on behalf of

Dr. Stefan Cristian Gherghina 

Academic Editor

PLOS ONE